# Multiplexed functional genomic analysis of 5′ untranslated region mutations across the spectrum of prostate cancer

Yiting Lim [1], Sonali Arora [1], Samantha L. Schuster [1,2], Lukas Corey[1], Matthew Fitzgibbon[3], Cynthia L. Wladyka[1], Xiaoying Wu[4], Ilsa M. Coleman[1], Jeffrey J. Delrow [3], Eva Corey [5], Lawrence D. True[5,6], Peter S. Nelson [1,7], Gavin Ha [8] & Andrew C. Hsieh [1,7 ✉]

The functional consequences of genetic variants within 5′ untranslated regions (UTRs) on a genome-wide scale are poorly understood in disease. Here we develop a high-throughput multi-layer functional genomics method called PLUMAGE (Pooled full-length UTR Multiplex Assay on Gene Expression) to quantify the molecular consequences of somatic 5′ UTR mutations in human prostate cancer. We show that 5′ UTR mutations can control transcript levels and mRNA translation rates through the creation of DNA binding elements or RNA-based *cis*-regulatory motifs. We discover that point mutations can simultaneously impact transcript and translation levels of the same gene. We provide evidence that functional 5′ UTR mutations in the MAP kinase signaling pathway can upregulate pathway-specific gene expression and are associated with clinical outcomes. Our study reveals the diverse mechanisms by which the mutational landscape of 5′ UTRs can co-opt gene expression and demonstrates that single nucleotide alterations within 5′ UTRs are functional in cancer.

[1] Divisions of Human Biology and Clinical Research, Fred Hutchinson Cancer Research Center, Seattle, WA, USA. [2] Molecular and Cellular Biology Graduate Program, University of Washington, Seattle, WA, USA. [3] Genomics and Bioinformatics Shared Resource, Fred Hutchinson Cancer Research Center, Seattle, WA, USA. [4] Division of Basic Sciences, Fred Hutchinson Cancer Research Center, Seattle, WA, USA. [5] Department of Urology, University of Washington, Seattle, WA, USA. [6] Department of Laboratory Medicine and Pathology, University of Washington, Seattle, WA, USA. [7] Departments of Medicine and Genome Sciences, University of Washington, Seattle, WA, USA. [8] Division of Public Health Sciences, Fred Hutchinson Cancer Research Center, Seattle, WA, USA. ✉email: ahsieh@fredhutch.org

The 5′-untranslated region (5′-UTR) lies within the non-coding genome upstream of coding sequences and plays a pivotal role in regulating gene expression[1]. Encoded within 5′-UTR DNA sequences are numerous *cis*-regulatory elements that can interact with the transcriptional machinery to regulate messenger RNA (mRNA) abundance[2]. Furthermore, transcribed 5′-UTRs are composed of a variety of RNA-based regulatory elements including the 5′-cap structure, secondary structures, RNA-binding protein motifs, upstream open-reading frames (uORFs), internal ribosome entry sites, terminal oligo pyrimidine (TOP) tracts, and G-quadruplexes. These elements can alter the efficiency of mRNA translation[3]; some can also affect mRNA transcript levels via changes in stability or degradation[4]. Individual mutations and single-nucleotide polymorphisms in 5′-UTRs have been reported in cancers, including mutations in the 5′-UTRs of oncogenes and tumor suppressors such as *c-MYC* and *TP53*[5,6]. Furthermore, individual 5′-UTR mutations in cancer have functional consequences. For example, mutations in the 5′-UTR of the tumor suppressor *RB1* alter RNA conformation and mRNA translation in retinoblastoma[7], while mutations in the 5′-UTR of *BRCA1* in breast cancer patients reduce translation efficiency (TE)[8]. On a genome-wide scale, recent studies of large patient cohorts have identified recurrent somatic 5′-UTR mutations across a variety of cancers[9,10]. Moreover, it has been shown that the overall 5′-UTR mutational burden within a cancer may influence malignant phenotypes[11]. Despite evidence pointing to the importance of 5′-UTR mutations in cancer and gene expression dynamics, a systematic functional interrogation of leader sequence mutations at both the transcription and translational levels has yet to be undertaken.

Massively parallel reporter assays (MPRAs) have been employed to dissect the functional consequences of genetic variation in regulatory elements such as promoters and enhancers[12,13]. These high-throughput technologies have enabled the characterization of these genomic regions on transcriptional activities. This approach has also been used to study UTR elements and their effects on mRNA degradation and translation[14–17]. While powerful, these studies have been limited to the investigation of short genomic regions <200 bases in length. This is an important limitation because 5′-UTRs range from 18 to >3000 bases, and UTR length and sequence context can have dramatic implications on gene expression[18]. Moreover, no studies to date have determined the functional landscape of 5′-UTR mutations across cancer progression at both the transcript and translation levels simultaneously. Thus, current approaches lack the ability to mine the breadth of full-length 5′-UTR activity and the depth of its impact on multiple layers of gene expression. Therefore, there is an urgent need for innovations that can overcome these barriers to allow for the analysis of the functional cancer-associated 5′-UTR-ome.

In this work, we develop a high-throughput approach for multilayer functional genomics within full-length 5′-UTRs called PLUMAGE (Pooled full-length UTR Multiplex Assay on Gene Expression). By coupling long- and short-read sequencing technologies, we overcome the length restriction of traditional MPRAs. In addition, we precisely quantify the effects of patient-based somatic mutations on both mRNA transcript levels and mRNA translation efficiency (TE) simultaneously, thereby providing an opportunity to interrogate multiple layers of gene expression regulation in cancer. To this end, we functionally interrogate 5′-UTR mutations identified in 229 localized and metastatic prostate cancer patients using PLUMAGE for their impact on mRNA transcript and translation levels. We observe that 35% of 5′-UTR mutations altered the transcript levels or translation rates across the spectrum of prostate cancer. The gene expression changes are driven in part by the creation of promoter

elements or by the disruption of RNA-based *cis*-regulatory motifs. We also identify 5′-UTR mutations in MAP kinase signaling pathway genes that are associated with changes in pathway-specific gene expression, responsiveness to taxane-based chemotherapy, and the development of metastases. Our functional study of the landscape of 5′-UTR mutations in a human malignancy highlights the molecular implications of this non-coding space in cancer pathogenesis and reveals previously undiscovered nodes of oncogenic gene regulation. In addition, PLUMAGE provides a technological platform for functional genomics of 5′-UTRs that can be applied to most genetically driven diseases.

## Results

**Somatic 5′-UTR mutations impact transcript levels and mRNA translation in human prostate cancer.** Localized prostate cancer is a highly prevalent disease and can evolve into metastatic castration-resistant prostate cancer (mCRPC), which is uniformly lethal[19]. While DNA- and RNA-based studies of human tissues ranging from localized to metastatic prostate cancer have been reported, the majority have focused on distant DNA-based regulatory regions or protein-coding regions[20–22]. As such, little is known about the mutational landscape of the 5′-UTR across the spectrum of human prostate cancer. Furthermore, it is unknown if 5′-UTR mutations influence transcript levels or mRNA translation in tumor tissues. To address these questions, we searched for all somatic 5′-UTR single-nucleotide variants in a cohort of five primary mCRPC patient-derived xenografts (PDXs) belonging to the LuCaP series[23], which encompass major genomic and phenotypic features of human prostate cancer, including adenocarcinoma (LuCaP 78, LuCaP 81, LuCaP 92), neuroendocrine prostate cancer (LuCaP 145.2), and a hypermutated prostate cancer (LuCaP 147). We observed a total of 326 mutations across all five PDXs, with the majority coming from LuCaP 147 (Supplementary Data 1a). These mutations did not localize to a particular region of the 5′-UTR relative to the ATG start codon (Fig. 1a) and were found in 5′-UTRs ranging from 42 to 2960 bases in length (Supplementary Fig. 1), suggesting the importance of assaying these mutations in the context of full-length 5′-UTRs. To determine if these 5′-UTR mutations were associated with changes in gene expression, we compared biological replicates of each LuCaP xenograft to a cohort of normal prostate tissues of high glandularity using human tissue-based transcriptome analysis and ribosome profiling (a whole-genome method to obtain a global snapshot of all translating mRNAs) (Supplementary Figs. 2 and 3a and Supplementary Data 1b, c)[24,25]. We observed high reproducibility between replicates, and successful capture of ribosome-bound fragments (Supplementary Fig. 3b–f). Our analyses uncovered that out of a total of 326 5′-UTR mutations in five LuCaP xenografts, 12–40% of mutations per xenograft were associated with an increase or decrease in transcript levels (false discovery rate (FDR) < 0.1, Fig. 1b). Similarly, at the level of translation, 15–38% of 5′-UTR mutations per xenograft were significantly associated with alterations in mRNA TE (FDR < 0.1, Fig. 1b). In total, 37% of 5′-UTR mutations had a positive or negative impact on transcript levels or translation in a mutually exclusive manner, suggesting multilayer regulation of gene expression. To reinforce our findings that the gene expression changes were attributed to the single-nucleotide mutations and not copy number alterations, we further conducted copy number analysis of all mutated loci and found that 33% of all copy neutral mutations also showed significant changes in gene expression. Given that our tissue-based studies only provide an association between 5′-UTR mutations and gene expression changes, and that these changes could be caused by other types of genetic

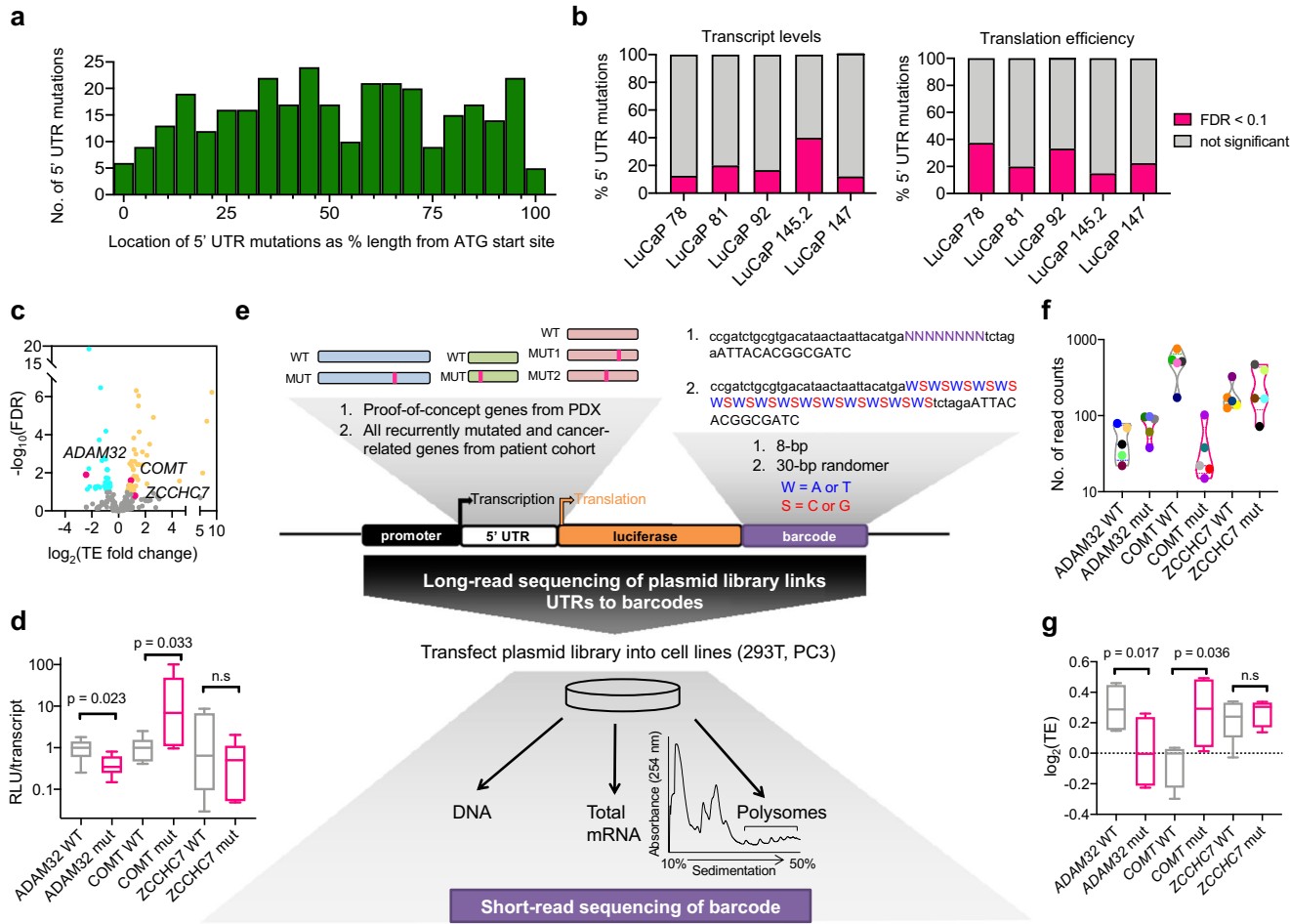

**Fig. 1 Development of a massively parallel reporter assay to quantify the impact of 5′-UTR mutations on transcript levels and mRNA translation.** **a** Histogram of the genomic distribution of all somatic single-nucleotide 5′-UTR mutations in five prostate cancer patient-derived xenografts (PDX) from the LuCaP series. **b** Percentage of 5′-UTR mutations in each LuCaP PDX that significantly alter transcript or mRNA translation efficiency (TE) levels (FDR < 0.1, represented in pink). Across five xenografts representing a spectrum of advanced human prostate cancer (adenocarcinoma, neuroendocrine prostate cancer, hypermutated prostate cancer), 13 mutations exhibited a decrease in transcript levels, while 35 mutations exhibited an increase. At the level of translation, 31 5′-UTR mutations decreased ribosome occupancy (decreased translation efficiency [TE]), while 42 had the opposite effect, independent of changes at the mRNA level. **c** Volcano plot showing TE fold changes of all 5′-UTR mutations in these LuCaP PDXs. Each dot represents TE fold change of a 5′-UTR mutation; turquoise dots are 5′-UTR mutations that significantly downregulate TE of its specific mRNA (FDR < 0.1), yellow dots are 5′-UTR mutations that significantly upregulate TE of its specific mRNA (FDR < 0.1). Thirty-one 5′-UTR mutations decreased ribosome occupancy (decreased translation efficiency [TE]), while 42 mutations increased TE. Mutations selected for orthogonal validation are demarcated in pink and labeled with the gene name. **d** Luciferase assays validating potentially functional 5′-UTR mutations (pink) identified by ribosome profiling including *ADAM32* (chr8: 38965236, C -> T) and *COMT* (chr22: 19939057, G -> A), as well as the negative control *ZCCHC7* (chr9: 37120713, C -> T). Normalization was performed by taking the ratio of the relative luminescence unit (RLU) to the amount of luciferase transcript determined by qPCR (n = 4–8 biological replicates, one-sided Student's t test). Data are presented as median by the center line, and the first and third quartiles as the upper and lower edges of the box. All minimum and maximum data points are indicated by error bars. **e** Simplified schematic of the Pooled full-length UTR Multiplex Assay on Gene Expression (PLUMAGE). **f** All 30 unique 8-bp barcodes were detected and linked with their respective WT and mutant 5′-UTR by PacBio long-read sequencing (average of 39.4–254.2 read counts per 5′-UTR–barcode pair). Each colored circle represents a unique 8-bp barcode. Data are presented as median by the center line. **g** Comparison of mRNA translation efficiency between WT and mutant *ADAM32*, *COMT*, and *ZCCHC7* 5′-UTRs by PLUMAGE. Mutations are represented in pink. Results are concordant with ribosome profiling and luciferase assay findings in (**c**, **d**). Normalized polysome read counts for each barcode per construct were taken as a ratio over normalized total RNA read counts for the same barcode (n = 4–5 biological replicates, one-sided Student's t test). Data are presented as median by the center line, and the first and third quartiles as the upper and lower edges of the box. All minimum and maximum data points are indicated by error bars. n.s. not statistically significant. Source data are provided as a Source data file.

alterations or noncancer cell types, we next sought to functionally test the ability of specific 5′-UTR mutations to alter protein abundance using an orthogonal luciferase assay. We focused on mutations in *ADAM32* (chr8: 38965236, C -> T), *COMT* (chr22: 19939057, G -> A), and *ZCCHC7* (chr9: 37120713, C -> T), which decreased, increased, or had no effect on mRNA translation, respectively. Luciferase assays of these 5′-UTR mutations confirmed our observations from ribosome profiling (Fig. 1c, d). Given the ability of the LuCaP xenografts to capture the diverse molecular composition of advanced human prostate cancer, these data support the concept that a subset of 5′-UTR mutations may alter gene expression in human tissue. Furthermore, the consequences of these mutations can be observed at both the transcript and translation levels in human-derived tumors.

**Development of an MPRA to quantify the impact of 5′-UTR mutations on transcript levels and mRNA translation efficiency**. Given the implications of these findings on our understanding of multilevel gene regulation in cancer, we sought to expand our functional investigation of 5′-UTR mutations genome-wide in a larger cohort of prostate cancer patients. Although ribosome profiling is a powerful method, it is laborious and low throughput, and would be challenging to implement on hundreds of patient samples. Furthermore, chromosomal alterations such as copy number changes and cell-type heterogeneity would make it difficult to definitively infer causality to single point mutations within the 5′-UTR. MPRAs are high-throughput technologies that enable the analysis of transcriptional or translational activities of myriad regulatory elements while controlling for gene dosage. However, historically they have suffered from two significant limitations. First, current MPRAs used to study 5′-UTR functionality are limited to the examination of short regions (50–125 bases) of the UTR-ome[15]. This is problematic because human 5′-UTRs can be as long as 3000 bases in length and mutations can occur anywhere along their length (Fig. 1a). Second, current MPRA technologies have yet to assay variants based on human cancers and show how such disease variants can regulate both transcript abundance and translation rates.

To overcome these limitations, we developed a method to assess the effects of prostate cancer patient-based somatic 5′-UTR mutations on mRNA transcript levels and mRNA translation rates in parallel, within the context of each full-length 5′-UTR (Fig. 1e). As a proof of principle, we cloned a small library of full-length wild-type (WT) and mutant 5′-UTRs from *ADAM32*, *COMT*, and *ZCCHC7* (Fig. 1c, d). We also included five unique 8-base pair (bp) barcodes per UTR variant at the 3′ end of the luciferase protein-coding sequence (CDS) (Supplementary Data 2). To quantify all 5′-UTR–barcode pairs within the library, we conducted long-read sequencing, which identified all plasmids with correct full-length 5′-UTR sequences properly linked to their corresponding unique barcodes (Fig. 1f). We transfected this library into PC3 prostate cancer cells, and 24 h later, extracted DNA, total mRNA, and polysome-bound mRNA (actively translating mRNA possessing three or more ribosomes) for short-read barcode sequencing. We observed that biological replicates were highly reproducible across all samples (Supplementary Fig. 4a). We found that the *ADAM32* 5′-UTR mutation (chr8: 38965236, C -> T) decreased, while the *COMT* 5′-UTR mutation (chr22: 19939057, G -> A) increased mRNA TE. The *ZCCHC7* 5′-UTR (chr9: 37120713, C -> T) mutation had no effect (Fig. 1g). These results replicated what we observed through orthogonal tissue-based ribosome profiling and luciferase assays (Fig. 1c, d). To further validate the ability of our approach to quantify changes in gene expression, we deleted the Kozak sequence and ATG codon of the luciferase gene and observed a decrease in TE (Supplementary Fig. 4b, c). We named this methodology PLUMAGE, and propose that it is a long- and short-read-linked sequencing approach that enables the massively parallel study of full-length 5′-UTRs and patient-based mutations on gene regulation at the mRNA transcript and translation levels.

**The 5′-UTR mutational landscape of human prostate cancer progression**. To interrogate 5′-UTR mutations encompassing both localized and metastatic prostate cancer in a large patient cohort, we analyzed existing whole-genome sequencing data from 149 localized prostate cancer patients[20,21] and supplemented this cohort with newly generated UTR sequencing of 80 end-stage mCRPC tumors. Collectively, we identified 2200 somatic single-nucleotide variants across 1878 genes (Supplementary Fig. 5a–c and Supplementary Data 3 and 4). Localized prostate cancers exhibited an average of 0.94 5′-UTR mutations per megabase

(mut/Mb), while mCRPCs averaged 1.46 mut/Mb ($p < 0.001$, Fig. 2a), which is consistent with mutation rates we observed in the protein-coding sequences (CDS) of these patients ($p < 0.001$, Supplementary Fig. 5d). Of the genes with 5′-UTR mutations, 45% also had alterations in the corresponding coding sequence. However, 55% of 5′-UTR mutations affected genes with no mutations in the protein-coding region ($p < 0.001$, Supplementary Fig. 5e). Interestingly, we observed that 246 genes with 5′-UTR mutations overlapped with the Quigley et al. mCRPC WGS dataset[26] (Supplementary Fig. 5f). We hypothesized that 5′-UTR-specific mutations may affect gene targets that are distinct from those observed in the CDS. To this end, gene set enrichment analysis (GSEA) revealed that 5′-UTR mutations more frequently impacted genes in the MAP kinase signaling and cell cycle pathways, while CDS mutations tend to be enriched in genes involved in cell adhesion processes (Fig. 2b). At a gene-specific level, *NOTCH2*[27], *FTH1*[28], and *CDH12*[29] harbored 5′-UTR mutations and have previously been implicated in prostate cancer pathogenesis (Supplementary Data 4e). Interestingly, other oncogenic factors not previously implicated in prostate cancer also exhibited 5′-UTR mutations including *MECOM* and *MLF1* (Supplementary Data 4e). Together, these findings demonstrate that somatic variants in the 5′-UTR genome appear to affect a diverse group of genes that may be functionally distinct from those that harbor CDS mutations.

**5′-UTR mutations are enriched in DNA- and RNA-binding motifs**. Given the diverse array of gene-specific molecular processes that the 5′-UTR regulates, including transcription and mRNA translation, we reasoned that somatic mutations within the 5′-UTR may be enriched in DNA and mRNA *cis*-regulatory regions. First, we analyzed the genomic locations of recurrently mutated 5′-UTRs found in two or more patients and observed that 38.7% of alterations are located within 50 bp of each other (Fig. 2c), suggesting that these clusters of mutations may be targeting functional elements[30]. Using the HOMER database for DNA-binding elements[31] and the Hughes database for human RNA-binding protein sites[32], we determined that 311 alterations mutated or created new known DNA-binding elements, and 478 alterations similarly affected known RNA-binding protein sites (Fig. 2d, e and Supplementary Data 5a, b). At the DNA level, the basic helix–loop–helix motif was the most frequently mutated DNA *cis*-element, while the *HuR*-, *SRSF1*-, and *TIA1*-binding motifs were the most frequently mutated RNA-binding sites (Supplementary Fig. 6). Next, we sought to determine if these observed mutations within *cis*-regulatory elements of 5′-UTRs occur more than would be expected by chance. We generated a background mutation distribution by randomly placing the equivalent number of 5′-UTR mutations we found in our analysis into the 5′-UTR-ome, taking into consideration the trinucleotide context of each mutation. This process was simulated 10,000 times. We observed that mutations that affect DNA and mRNA motifs within the 5′-UTR region of the genome occur more frequently in prostate cancer patients than would be expected by chance (Fig. 2d). Lastly, we queried specific known translation regulatory elements such as uORFs, TOP (terminal oligo pyrimidine)-like/PRTE (pyrimidine-rich translational element) motifs, G-quadruplexes, and 5′-TOP motifs to determine if somatic mutations are evenly distributed across motifs in general. We observed that amongst these known 5′-UTR *cis*-elements, only TOP-like/PRTE motifs exhibited statistically significant enrichment of mutations ($p < 0.001$, Fig. 2e). Overall, these data show that 5′-UTR mutations are present in DNA and RNA *cis*-elements, suggesting that they may functionally impact mRNA transcript levels and mRNA translation.

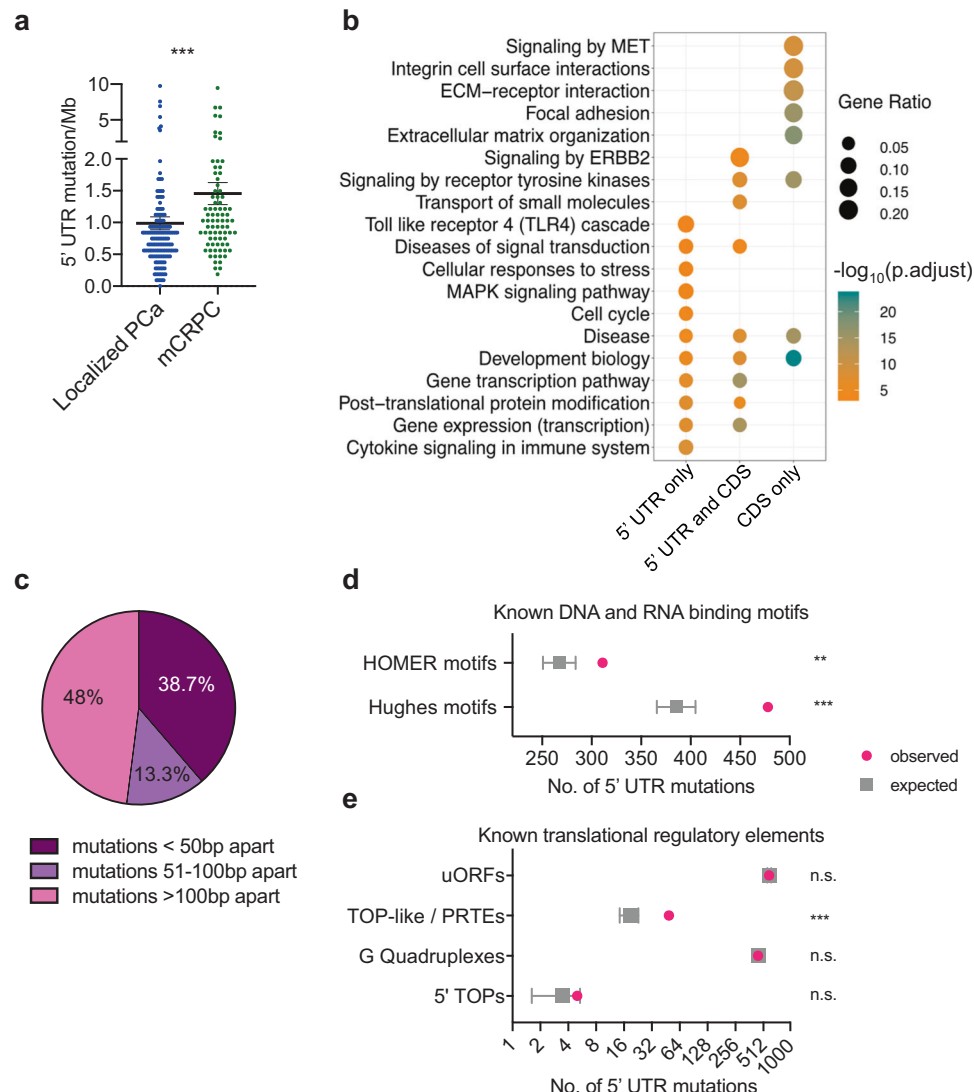

**Fig. 2 The mutational landscape of prostate cancer 5′-UTRs. a** Comparison of 5′-UTR mutation rate (5′-UTR mutation/Mb) in localized prostate cancer (PCa) patients ($n = 149$, each patient represented by a blue dot) and metastatic castration-resistant prostate cancer (mCRPC) patients ($n = 80$, each patient represented by a green dot). Each dot represents the mutation rate per patient (***$p = 0.0001$, two-tailed Mann–Whitney $U$ test). Data are presented as mean values ± s.e.m. **b** KEGG and Reactome pathway analyses of all genes with 5′-UTR and protein-coding sequence (CDS) mutations across 229 prostate cancer patients. Genes with 5′-UTR mutations can cluster with or independent of genes with CDS mutations (Fisher's hypergeometric test, FDR < 0.05). **c** The absolute genomic distance of somatic single-nucleotide 5′-UTR mutations within recurrently mutated genes; 38.7% of recurrently mutated 5′-UTRs have alterations located less than 50-bp apart. **d** Predicted enrichment of observed 5′-UTR mutations in our patient cohort across known DNA- and RNA-binding regulatory elements. Validated DNA (Homer) and RNA protein-binding motifs (Hughes) were analyzed. To generate the background (null) distribution of mutations, permutations of all 5′-UTR mutation locations ($n = 2200$ mutations) found in our dataset were performed ~10,000 times taking into account covariates such as trinucleotide context. The total number of observed mutations (represented by pink dots) impacting each regulatory element type was compared to the background distribution of the permutation data (represented by gray square) and the $p$ value was computed from a distribution obtained from a Monte Carlo simulation (**$p = 0.001$, ***$p = 0.0001$). Data are presented as mean values ± s.d. **e** Predicted enrichment of observed 5′-UTR mutations in our patient cohort across *cis*-regulatory elements known to affect translation such as upstream open-reading frames (uORFs), terminal oligo pyrimidine (TOP)-like or pyrimidine-rich translational elements (PRTEs), G-quadruplexes, and 5′-TOP elements. To generate the background (null) distribution of mutations, permutations of all 5′-UTR mutation locations ($n = 2200$ mutations, represented by pink dots) found in our dataset were performed ~10,000 times taking into account covariates such as trinucleotide context. The total number of observed mutations impacting each regulatory element type was compared to the background distribution of the permutation data (represented by gray square) and the $p$ value was computed from a distribution obtained from a Monte Carlo simulation (***$p = 0.0001$). Data are presented as mean values ± s.d. n.s. not statistically significant. Source data are provided as a Source data file.

**Functional analysis of 5′-UTR mutations across the spectrum of human prostate cancer**. To comprehensively assay the functional landscape of 5′-UTR mutations, we developed a second PLUMAGE library. This larger library was composed of 914 synthesized full-length 5′-UTR sequences covering 545 somatic mutations from all recurrent (two or more patients)

and cancer-associated 5′-UTR mutations identified in 229 patients (Fig. 1e, Supplementary Fig. 7a, and Supplementary Data 6a). Here, instead of using five 8-bp barcodes per 5′-UTR, a 30-bp randomer barcode was cloned downstream of the luciferase CDS. We constrained the library to 212,325 unique barcodes with an average of 236 barcodes per 5′-UTR, which enabled deep

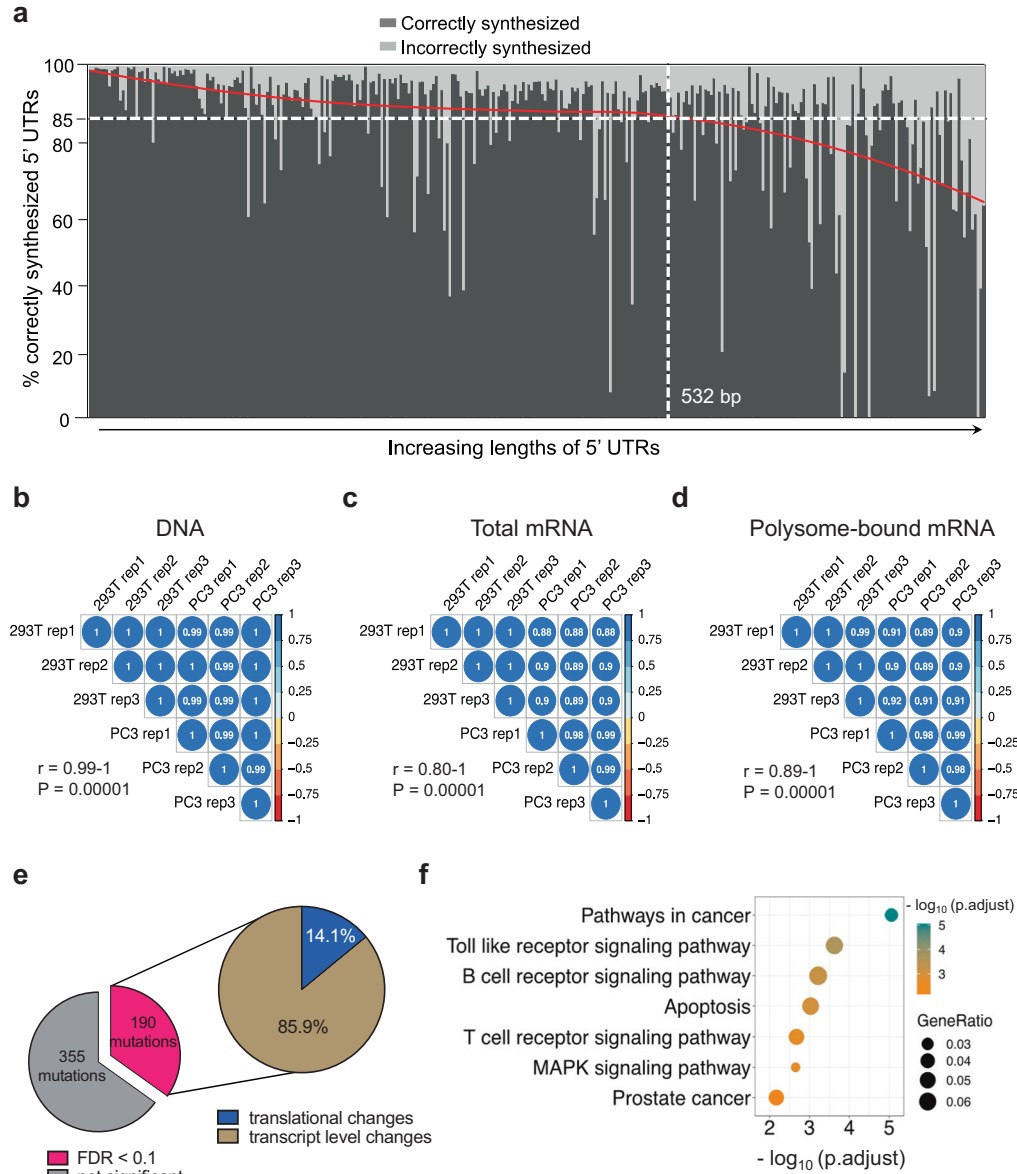

**Fig. 3 Thirty-five percent of 5′-UTR mutations across the spectrum of human prostate cancer functionally impact transcription or translation. a** Per-gene percentages of distinct barcodes associated with an exact match to an expected 5′-UTR sequence by PacBio long-read sequencing. All distinct 5′-UTR sequences were observed by long-read sequencing and linked to an average of 236 distinct 30-bp barcodes. For each gene, the percentage of barcodes associated with an exactly matching 5′-UTR are plotted as black vertical bars (correctly synthesized). Genes are ordered by increasing 5′-UTR length from left to right, and the average rate of exactly matching barcodes is marked by a horizontal dashed line at 85%. A smoothed fit, using loess regression, of percentage matching vs rank order of length is shown in red. **b** Correlation of normalized read counts per WT and mutated 5′-UTR in each technical and biological replicate for each PLUMAGE DNA sample. Three biological replicates were analyzed for each cell line (293T and PC3). Pearson correlation coefficient was calculated to determine significance and was found to be $r > 0.99$ for all samples (all $p$ values = 0.00001). **c** Correlation of normalized read counts per WT and mutated 5′-UTR in each technical and biological replicate for each PLUMAGE total mRNA sample. Three biological replicates were analyzed for each cell line (293T and PC3). Pearson correlation coefficient was calculated to determine significance and was found to be $r > 0.8$ for all samples (all $p$ values = 0.00001). **d** Correlation of normalized read counts per WT and mutated 5′-UTR in each technical and biological replicate for each PLUMAGE polysome-bound mRNA sample. Three biological replicates were analyzed for each cell line (293T and PC3). Pearson correlation coefficient was calculated to determine significance and was found to be $r > 0.89$ for all samples (all $p$ values < 0.0001). **e** Proportion of all 5′-UTR mutations assayed by PLUMAGE that showed a significant (FDR < 0.1, in pink) change in mRNA transcript or translation levels. **f** 5′-UTR mutations (190 mutations) that significantly change gene expression affect important cancer-related pathways by KEGG pathway analysis (Fisher's hypergeometric test, FDR < 0.05). Source data are provided as a Source data file.

sampling of each mutated and unmutated 5′-UTR. To determine the 5′-UTR–barcode pair identities and ensure the analysis of only correctly synthesized 5′-UTRs, we conducted long-range sequencing of our entire plasmid library. Here, we observed that 85% of sequenced plasmids had the correct WT or mutant 5′-UTR sequences. Interestingly, 89.9% of 5′-UTR sequences shorter than 532 bases were correctly synthesized, while only 77.1% of sequences >532 bases were correctly synthesized (Fig. 3a). Thus,

our highly saturated barcode system is essential to identify correctly synthesized constructs for further downstream analysis and provides a solution to confidently assay 5′-UTRs of any length.

The plasmid library was transfected into human PC3 prostate cancer cells and human embryonic kidney 293T cells. After 24 h, DNA, total mRNA, and polysome-bound mRNA (mRNA associated with three or more ribosomes) were isolated and sequenced (Supplementary Fig. 7b and Supplementary Data 6b). Short-read sequencing of the 30-bp barcodes in each DNA, total mRNA, and polysome-bound mRNA sample showed a strong correlation across three biological replicates in both cell lines (Fig. 3b–d). After filtering for incorrectly synthesized or cloned constructs using the long-read dataset (Fig. 3a and Supplementary Data 6a), each WT and mutant full-length 5′-UTR was represented by an average of 214 unique 30-bp barcodes (minimum normalized read count of 0.5 counts per million, Supplementary Fig. 8a, b and Supplementary Data 6c). Notably, all constructs reliably detected by long-read sequencing were identified in all DNA, total mRNA, and polysome-bound mRNA samples by short-read sequencing. Furthermore, we observed a strong correlation between 293T and PC3 cells across all replicates in all samples, suggesting high reproducibility across different cell lines (Pearson $r > 0.8$, $p < 0.0001$, Fig. 3b–d). Small differences in correlation were primarily seen between cell lines, suggesting that some mutations may have a context-specific dependency. To quantitate changes in mRNA transcript levels and control for gene dosage, we normalized the total mRNA barcode read counts to the corresponding DNA read counts of that particular barcode, and measured fold change by comparing the mutant 5′-UTR to the WT 5′-UTR. To quantitate changes in mRNA translation, we measured TE, which we determined by calculating the ratio of normalized polysome-bound mRNA barcodes to their respective total mRNA barcodes, and similarly measured fold change by comparing the mutant 5′-UTR to the WT 5′-UTR. Out of a total of 545 mutations assayed, 190 showed significant changes in mRNA transcript levels or mRNA TE (FDR < 0.1, Fig. 3e and Supplementary Data 6d, e). To ensure that changes at the level of translation were reflective of differential ribosome loading, we also measured the polysome to 80S ratio, which highly correlated with changes in TE (Pearson $r = 0.768$, $p = 0.0001$, Supplementary Fig. 9a). Importantly, these functional mutations affected genes that belong to cancer-related pathways and include the oncogenic *FTH1* and tumor-suppressive *MECOM* (Fig. 3f and Supplementary Data 6d). It has been previously reported that germline variants within the 5′-UTR that create *cis*-regulatory elements, such as uORFs, are under strong negative selection[33]. As such, we asked if our positive PLUMAGE hits were associated with genomic structural variations within the 5′-UTR compared to 5′-UTR mutations that were not functional. Comparing the localized copy numbers of the mutant 5′-UTR on a per-patient basis, we observed no increase in copy number losses as a result of having a functional 5′-UTR mutation (Supplementary Fig. 9b). Together, these findings demonstrate that 34.8% of mutated 5′-UTRs harbor functional alterations that can deregulate cancer-related genes in prostate malignancies at either the transcriptional or translational levels.

**Orthogonal validation of PLUMAGE reveals functional 5′-UTR mutations that create neo-promoters, disrupt RNA *cis*-elements, or affect multilevel gene regulation.** To validate functional 5′-UTR mutations identified through PLUMAGE, we tested individual WT and mutant pairs by orthogonal quantitative PCR (qPCR) and luciferase assays. Mutations that impact transcript levels were found in oncogenic genes such as *FOS* (chr14: 75745674, C -> G) and *FGF7* (chr15: 49715462, C -> T),

which are components of the MAP kinase signaling pathway and known to drive prostate cancer pathogenesis (Fig. 4a)[34,35]. A recent study of mCRPC patients reported that MAP kinase and fibroblast growth factor (FGF) signaling pathways are active and promote growth in a subtype of highly aggressive mCRPC[36]. However, the mechanism for this activation remains elusive. We reasoned that these 5′-UTR mutations could impact the expression levels of critical MAP kinase pathway components. Indeed, orthogonal validation of the *FOS* and *FGF7* 5′-UTR mutations in human prostate cancer cells revealed that these single-nucleotide alterations identified through PLUMAGE were sufficient to increase gene-specific transcript levels (Fig. 4b). Importantly, the increase in mRNA transcript levels was not caused by impaired mRNA degradation due to the mutation or by the different 30-bp barcode used between the WT and mutant constructs (Supplementary Fig. 10a, b).

Next, we sought to determine if this increase in transcript could be observed in human tissues. Interestingly, the *FGF7* (chr15: 49715462, C -> T) 5′-UTR alteration present in a PDX specimen was associated with a significant increase in *FGF7* mRNA transcript abundance by a $\log_2$ fold change of 3.09 (Fig. 4c). Furthermore, we noticed that this specific mutation created a new DNA-binding site element (Fig. 4d). In particular, the somatic alteration from CACGCG to CACGTG created an E-box, the canonical binding motif for the oncogenic MYC protein, which is often deregulated in advanced prostate cancer patients[37]. Importantly, it has been shown that MYC can bind to E-box elements downstream of the transcriptional start site to promote gene-specific expression[38]. To verify binding, we performed an electrophoretic mobility shift assay (EMSA) and found that the MYC:MAX heterodimer protein complex was specifically bound to the E-box sequence created in the mutated *FGF7* 5′-UTR, but did not bind the WT sequence (Fig. 4e). Heterodimer binding was also abolished in the presence of an unlabeled oligonucleotide competitor and when MYC and MAX were tested individually, suggesting specific affinity for the E-box sequence created by the *FGF7* 5′-UTR mutation (Fig. 4e). These findings reaffirm the 5′-UTR as a dynamic region containing regulatory elements that impact transcript levels and demonstrate that the creation of neo-promoter elements can affect transcription factor binding and mRNA transcript levels of oncogenic factors.

Interestingly, 57.8% of mutations that affected mRNA translation also changed a putative RNA-based *cis*-regulatory element (Fig. 5a). Such mutations were found in mRNAs that included the oncogene *AKT3*[39], the microtubule-binding protein *NUMA1*, which also has tumor-suppressive properties[40], and the oncogenic cyclin-dependent kinases regulatory subunit *CKS2*[41] (Fig. 5a). Using luciferase assay normalized by gene-specific transcript levels, we determined the *AKT3* mutation (chr1: 244006547, C -> T) indeed leads to an increase in protein levels, whereas the *NUMA1* mutation (chr11: 71780891, C -> A) decreases protein abundance (Fig. 5b). Importantly, the increase and decrease in protein abundance were not affected by the 30-mer barcode encoded in each luciferase construct (Supplementary Fig. 11a, b). Interestingly, we found that the *NUMA1* mutation removes a serine- and arginine-rich splicing factor 9 (SRSF9) RNA-binding protein motif found in the Hughes and CISBP RNA-binding protein motif datasets (Fig. 5c and Supplementary Data 7). SRSF9, which is predicted to interact with this motif, has been implicated in tumorigenesis by deregulating the proper translation of specific mRNAs such as β-catenin[42]. This RNA-binding motif appears to be conserved in the serine- and arginine-rich protein family; thus, the abrogation of the motif may represent a larger node of gene regulation[43].

One of the unique features of PLUMAGE is the ability to monitor changes at both the transcript and mRNA translation

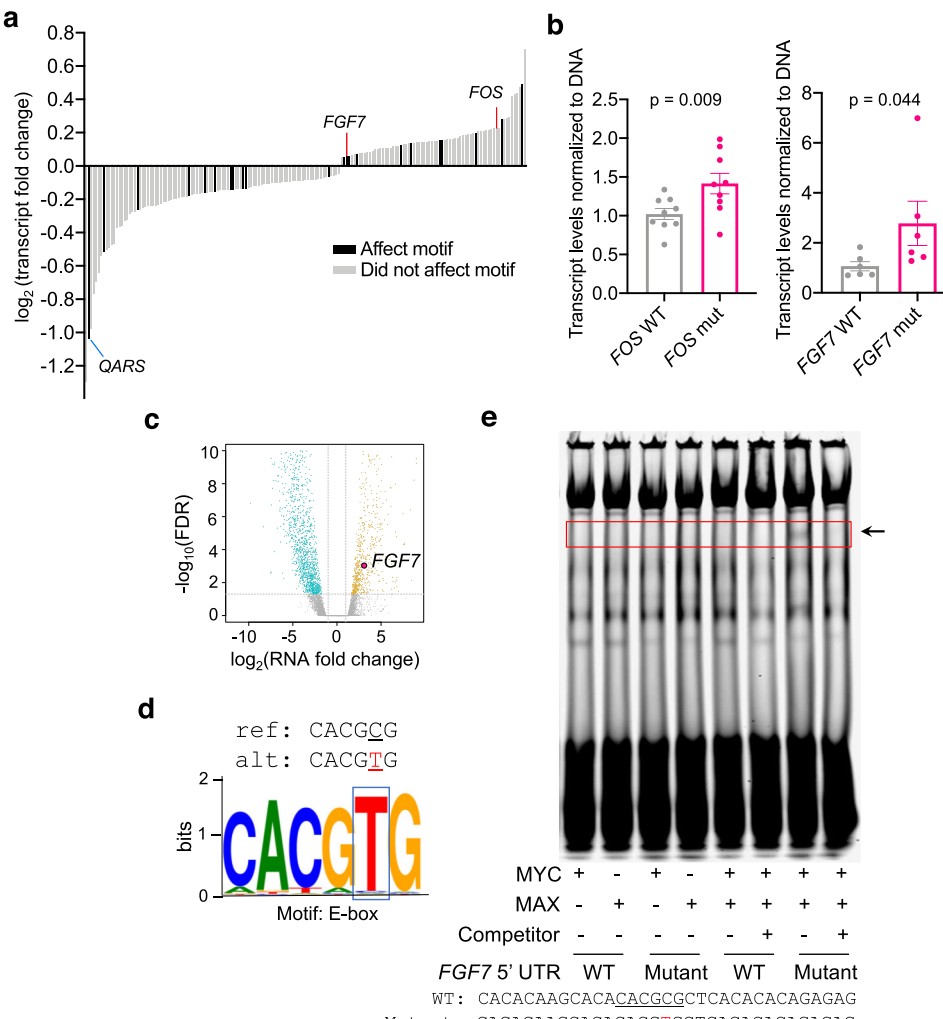

**Fig. 4 PLUMAGE reveals how somatic 5′-UTR mutations affect mRNA transcript levels. a** 5′-UTR mutations that significantly affect mRNA transcript levels (Mann–Whitney $U$ test, FDR < 0.1) and magnitude fold change compared to unmutated 5′-UTR (Supplementary Data 6d). A proportion of mutations also impact a known DNA-binding element, indicated by black bars. **b** qPCR validation of the *FOS* (chr14: 75745674, C -> G) and *FGF7* (chr15: 49715462, C -> T) 5′-UTR mutations identified by PLUMAGE. WT (represented by gray dots) and mutant (represented by pink dots) 5′-UTRs were cloned into a luciferase reporter construct and transduced into PC3 prostate cancer cells. Luciferase transcript levels were then normalized to luciferase DNA ($n = 9$ biological replicates for *FOS* WT and mutant, mean ± s.e.m. Student's $t$ test; $n = 6$ biological replicates for *FGF7* WT and mutant, data are presented as mean ± s.e.m., one-sided Student's $t$ test). **c** RNAseq volcano plot of all significantly up- and down-regulated mRNAs in the human prostate cancer PDX LuCaP 81 (FDR < 0.1). Each dot represents the transcript fold change of a 5′-UTR mutation in LuCaP 81; turquoise dots are 5′-UTR mutations that significantly downregulate transcript expression of its specific mRNA (FDR < 0.1), yellow dots are 5′-UTR mutations that significantly upregulate transcript expression of its specific mRNA (FDR < 0.1). Within this PDX, *FGF7* exhibits a 5′-UTR mutation (indicated by pink dot) at chr15: 49715462, C -> T that is associated with an increase in *FGF7* transcript levels. **d** The *FGF7* 5′-UTR mutation introduces a thymidine at position chr15: 49715462, which transforms the CACGCG sequence into an E-box motif (CACGTG). Mutated nucleotide is represented in red. **e** Representative EMSA using the WT vs mutant *FGF7* 5′-UTR. Labeled probe sequences (33-bp) containing the E-box sequence generated by the mutation in the 5′-UTR of *FGF7* and the wild-type sequence are shown. Mutated nucleotide is represented in red. Binding of MYC:MAX heterodimer protein complex is observed only with the mutated oligonucleotide probe containing the E-box sequence. Binding of the labeled oligo can be abolished using an unlabeled competitor. Source data are provided as a Source data file.

levels simultaneously. Indeed, we found that a single point mutation in the 5′-UTR of glutaminyl-tRNA synthetase (*QARS*) (chr3: 49142179, G -> A) led to a concomitant decrease at the transcript level, but an increase at the level of mRNA translation, which we validated by qPCR and luciferase assay (Fig. 5d). We confirmed that the differences we observed were not a result of differential transfection efficiency between the WT and mutant constructs (Fig. 5d). As such, PLUMAGE is able to uncover 5′-UTR mutations that transcend a single mode of gene expression. Furthermore, these data support the concept that 5′-UTR mutations found in cancer patients within a singular genomic

location can dictate the fate of specific genes by co-opting multiple levels of gene expression. Together, these findings demonstrate the significant utility of PLUMAGE in delineating new ways in which somatic mutations within the 5′-UTR can affect the molecular outcome of cancer-associated genes.

**CKS2 5′-UTR mutation engineered by CRISPR base editing increases endogenous translation.** We observed that a C -> T mutation in the 5′-UTR of oncogenic *CKS2* (chr9: 91926143) creates a new upstream AUG (uAUG) within the 5′-UTR in-frame with the main reading frame (Fig. 6a). Interestingly, this

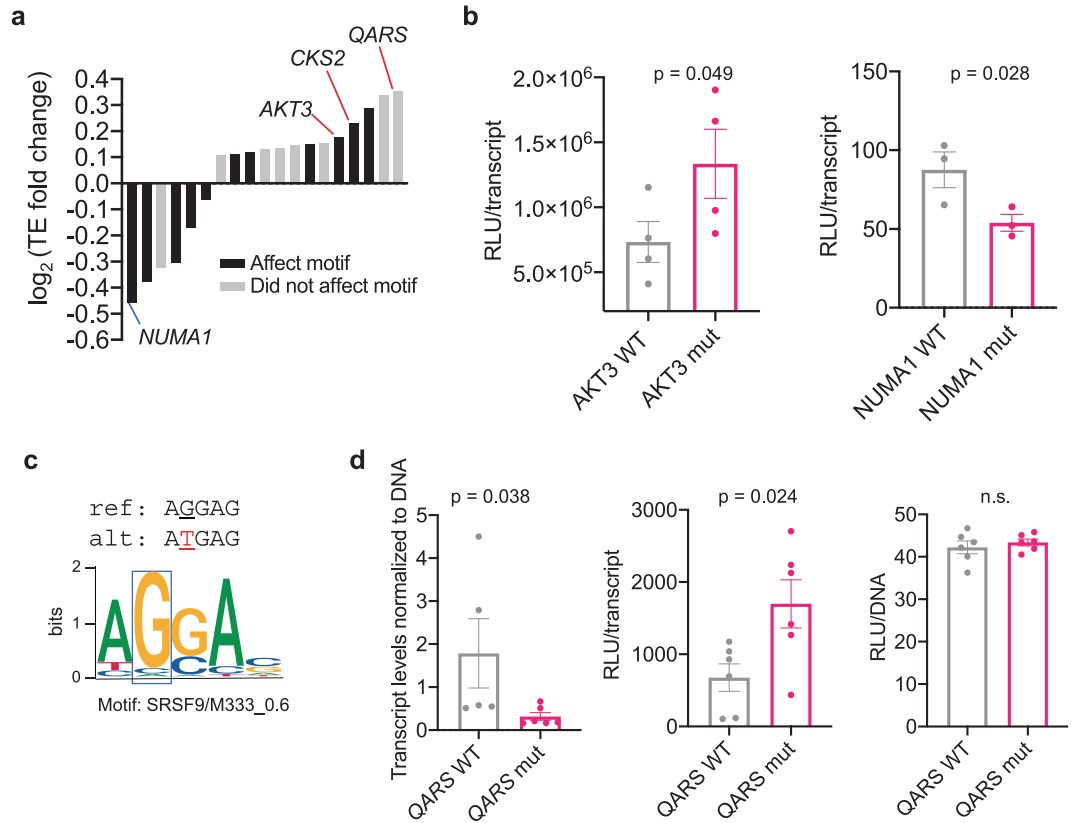

**Fig. 5 PLUMAGE uncovers somatic 5′-UTR mutations that affect mRNA translation efficiency and multiple layers of gene expression. a** 5′-UTR mutations that significantly affect mRNA translation efficiency (Mann–Whitney $U$ test, FDR < 0.1) and magnitude fold change compared to unmutated 5′-UTRs (Supplementary Data 6e). A proportion of mutations impact known RNA-binding protein-binding motifs, indicated by black bars. **b** Validation of 5′-UTR mutations in *AKT3* (chr1: 244006547, C -> T) and *NUMA1* (chr11: 71780891, C -> A) by luciferase assay. Each WT (represented in gray) or mutant (represented in pink) construct was separately transfected into PC3 cells and assayed for luciferase activity and luciferase mRNA expression after 24 h. Luciferase activity (RLU) was normalized to the amount of luciferase transcript in each transfection to determine translation efficiency ($n = 4$ biological replicates for *AKT3* WT and mutant, mean ± s.e.m. Student's $t$ test; $n = 3$ biological replicates for *NUMA1* WT and mutant, data are presented as mean ± s.e.m., one-sided Student's $t$ test). **c** The C -> A 5′-UTR mutation in *NUMA1* at position chr11: 71780891 abrogates an existing SRSF9 RNA-binding protein motif. Mutated nucleotide is represented in red. **d** The 5′-UTR mutation in *QARS* (chr3: 49142179, G -> A) shows significant changes in both transcript levels and translation efficiency, not attributable to the amount of DNA transfected. Each *QARS* 5′-UTR WT (represented in gray) and mutant (represented in pink) plasmid was transfected individually into PC3 cells ($n = 5$–6 biological replicates), followed by luciferase assay, luciferase RNA PCR, and luciferase DNA qPCR. RLU values were normalized to luciferase mRNA to determine translation efficiency. Luciferase mRNA was normalized to luciferase DNA to determine the effects on transcript levels ($n = 6$ biological replicates, data are presented as mean ± s.e.m., one-sided Student's $t$ test). n.s. not statistically significant. Source data are provided as a Source data file.

uAUG increased overall translation through the *CKS2* 5′-UTR in PLUMAGE (Fig. 5a). To determine whether this mutation would increase translation of endogenous *CKS2*, we engineered the C -> T mutation using CRISPR cytosine base editing. Nucleotide-specific base editing utilizes a complex of a cytosine deaminase (APOBEC1), a Cas9-nickase, and uracil-DNA glycosylase inhibitor (UGI) (Fig. 6b). The Cas9 domain directs the complex to a target locus, where APOBEC1 deaminates a cytosine to uracil. This results in a G–U base pair that is protected from excision by UGI, which is replaced by cellular mismatch repair to an A–T base pair in place of the original C–G[44]. Using a single guide RNA (sgRNA) targeting the *CKS2* 5′-UTR and the evolved evoAPOBEC1-BE4max-NG base editor, we successfully generated three *CKS2* 5′-UTR mutant cell lines, each with one allele possessing the C -> T mutation (Fig. 6c). Next, we measured CKS2 protein levels by western blot analysis and observed a 2-fold increase in overall CKS2 protein in mutant cells compared to WT controls (Fig. 6d). This increase was mainly due to the expression of the 14 kDa extended CKS2 coding sequence without any noticeable sacrifice of the normal 11 kDa CKS2 coding sequence. Importantly, the specificity of the CKS2 antibody for the 11 and

14 kDa N terminally extended CKS2 was confirmed by short hairpin RNA (shRNA) knockdown (Supplementary Fig. 11c). Since *CKS2* mRNA expression did not differ between the WT and mutant cells (Fig. 6e), we concluded that the *CKS2* chr9: 91926143 C -> T mutation was sufficient to increase mRNA-specific translation. This observation corroborates our PLUMAGE findings and demonstrates that 5′-UTR mutations can coordinately impact mRNA translation by altering RNA-based *cis*-regulatory elements in their endogenous context.

**5′-UTR mutations in MAP kinase regulators are associated with gene expression changes, response to taxane-based chemotherapy, and metastasis.** Our patient cohort consists of both localized prostate cancer and mCRPC patients, thus enabling the study of the impact of 5′-UTR mutations in early-stage versus advanced metastatic prostate cancer. We found 5′-UTR mutations that were unique to either localized cancer or mCRPC (Supplementary Data 4e). Indeed, GSEA analyses showed that 5′-UTR mutations in localized prostate cancer enrich for cell cycle pathways, whereas mutated genes in mCRPC enrich for

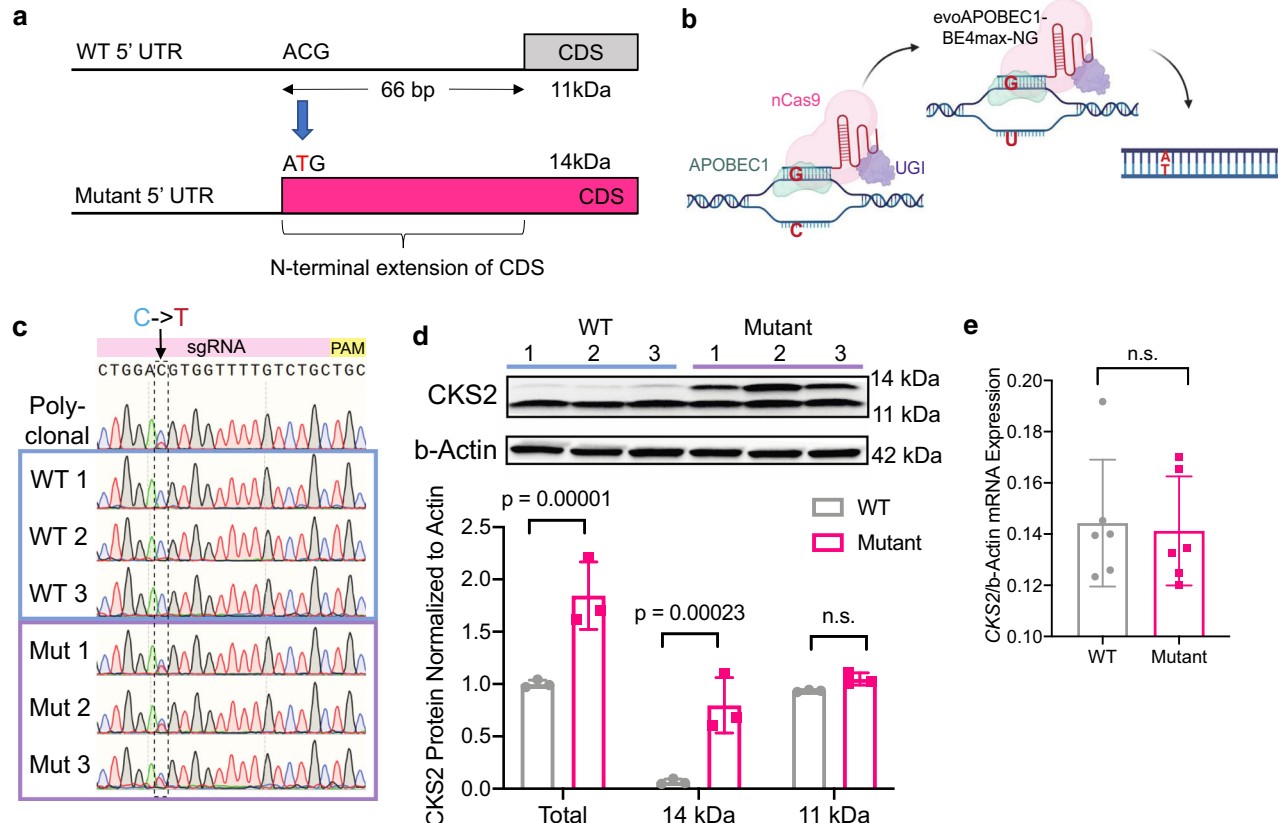

**Fig. 6 CRISPR-Cas9 base editing of a point mutation in the CKS2 5′-UTR increased translation efficiency in its endogenous context. a** Schematic showing wild-type (WT, top) and mutant (bottom) versions of *CKS2* transcript, including 5′-UTR, normal coding sequence (CDS), and mutant N terminally extended CDS. The C to T mutation (chr9: 91926143) within the 5′-UTR of *CKS2* generates a start codon that extends the coding sequence of *CKS2*. The mutated nucleotide is represented in red. **b** Method of CRISPR-Cas9 base editing using evoAPOBEC1-BE4max-NG, which is composed of APOBEC1, a Cas9-nickase domain, and uracil-DNA glycosylase inhibitor (UGI). This base editor deaminates target cytosines to uracil, which changes the original G–C base pair into an A–T base pair after DNA repair. **c** Sanger sequencing traces from polyclonal population of CRISPR-transfected 293T cells and six individual single-cell clones selected from this pool for further study. The target C (blue) -> T (red) mutation in the 5′-UTR of *CKS2* is shown within the dashed box. **d** Western blot of the three WT and three *CKS2* mutant clonal cell lines created by CRISPR base editing with antibodies against CKS2 and β-actin. The graph shows these results quantified using ImageJ, where each CKS2 band intensity was measured and normalized to the intensity of the corresponding β-actin loading control. Statistics show two-sided Student's *t* test with multiple comparisons correction using the three WT (gray) vs three *CKS2* mutant (pink) biological replicates (total *p* value = 0.00001 and 14 kDa *p* value = 0.00023). Data are presented as mean values ± s.d. Full immunoblots are provided in the Source data file. **e** *CKS2* qPCR shows no change in mRNA levels between three WT (represented in gray) and three mutant (represented in pink) clonal cell lines created from CRISPR base editing. *CKS2* mRNA levels in each sample were normalized to β-actin as a loading control (*n* = 6 biological replicates). Two-sided Student's *t* test, data are presented as mean values ± s.e.m. n.s. not statistically significant. Source data are provided as a Source data file.

metabolism and the MAP kinase signaling pathway (Fig. 7a). To determine if functional 5′-UTR mutations of MAP kinase regulators can impact pathway-specific gene expression, we analyzed RNA-sequencing (RNAseq) data from patients and PDX models harboring MAP kinase pathway mutations we tested in PLUMAGE. Interestingly, three patients with functional MAP kinase pathway 5′-UTR mutations to *FOS*, *FGF7*, and *MECOM* that were predicted to increase signaling by PLUMAGE demonstrated upregulation of a RAS-driven prostate cancer MAP kinase pathway gene signature[45] (Fig. 7b). Moreover, three patients with nonfunctional 5′-UTR mutations to MAP kinase pathway genes *PTPN7*, *JUN*, and *DDIT3* did not exhibit changes in gene expression (Fig. 7b). These findings demonstrate that functional 5′-UTR mutations have the ability to impact cancer-associated pathway activity.

Next, we sought to determine if functional mutations within 5′-UTRs of MAP kinase pathway regulators are associated with patient outcomes. We analyzed multiple patient endpoints including progression-free survival, overall survival, time to

metastases, Gleason score, and duration on therapeutic agents. Interestingly, we observed that patients with functional MAP kinase pathway mutations that were predicted to increase signaling by PLUMAGE (*FOS* and *MECOM*, FDR < 0.1) were more likely to have a sustained response to the microtubule inhibitor and chemotherapy Taxotere compared to patients without functional mutations (Supplementary Fig. 12). This observation is consistent with a previous report that demonstrated that taxane-mediated apoptosis is dependent on MAP kinase signaling[46]. While intriguing, these findings are limited by the small sample size of patients with functional MAP kinase pathway 5′-UTR mutations and larger studies will be required to determine the generalizability of these findings.

Lastly, we sought to determine if 5′-UTR mutations to MAP kinase regulators could represent a biomarker for disease aggressiveness because it has been implicated in prostate cancer metastasis[47]. To increase the power of this analysis, we asked if all 19 MAP kinase 5′-UTR mutations we observed in metastatic patients correlated with disease physiology (Fig. 7c). Notably,

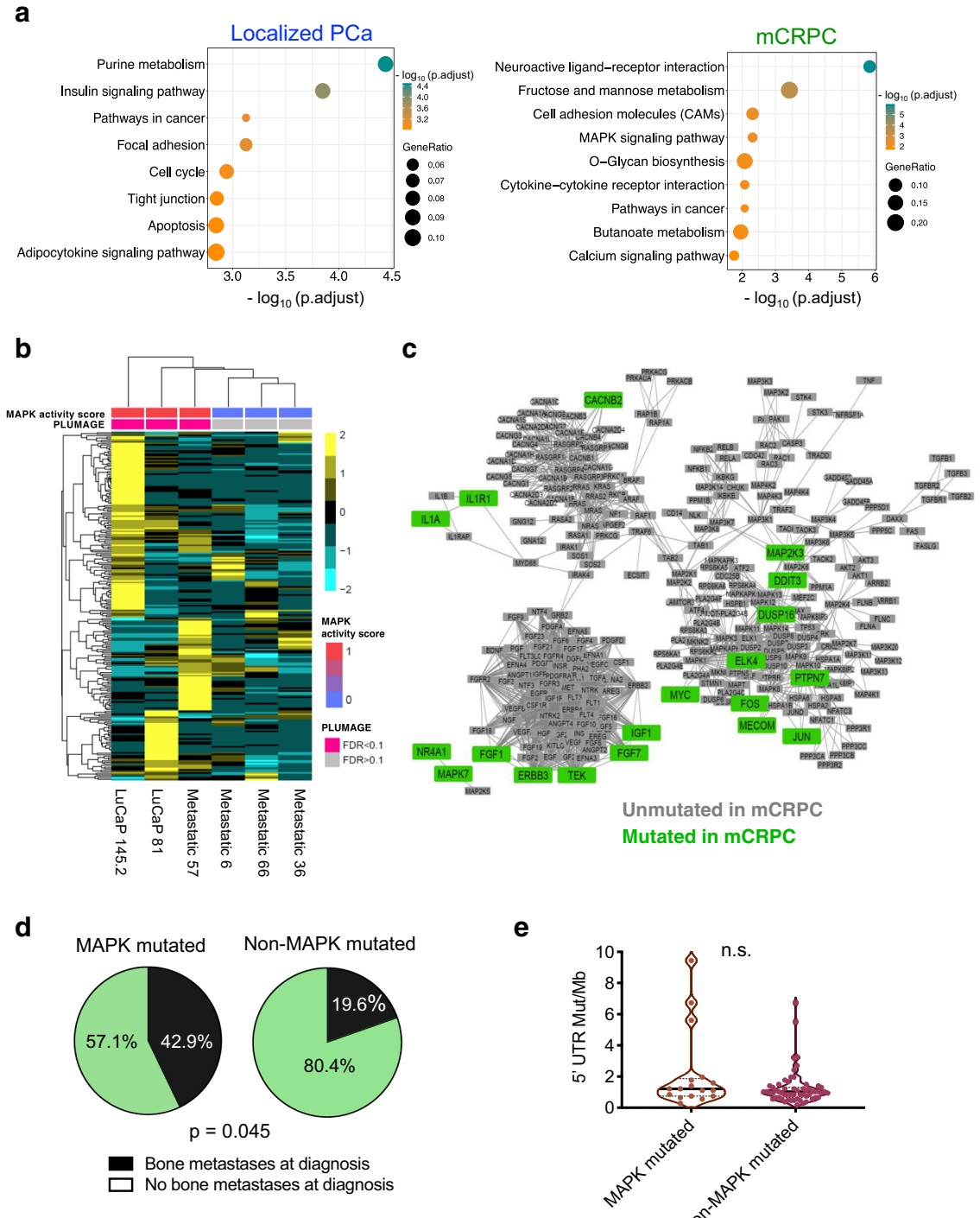

**Fig. 7 5'-UTR mutations in MAP kinase signaling pathway genes associated with increased pathway activity and metastases in prostate cancer patients. a** Genes with 5'-UTR mutations in localized and advanced prostate cancer cluster into distinct functional categories as determined by KEGG pathway analysis (Fisher's hypergeometric test, FDR < 0.05). **b** Heatmap of a MAP kinase pathway activity signature demonstrating that patients with functional 5'-UTR mutations to MAP kinase regulators (PLUMAGE FDR < 0.1) exhibit increased pathway activation compared to nonfunctional mutations (PLUMAGE FDR > 0.1). The heatmap color key represents normalized $\log_2(\text{CPM} + 1)$ values for each gene. **c** Metastatic castration-resistant prostate cancer patients harbor 5'-UTR mutations within genes found in the MAP kinase signaling pathway. Gene names in green represent those with 5'-UTR mutations in mCRPC patients. Gene names in gray are MAP kinase signaling pathway components and downstream effectors that are not mutated in mCRPC patients. **d** mCRPC patients with mutated MAP kinase pathway genes are significantly more prone to bone metastases at diagnosis (black) compared to patients who do not harbor these mutations (green) ($p = 0.045$, two-sided Student's $t$ test). **e** The difference in bone metastasis at diagnosis between the two patient groups is independent of any differences in 5'-UTR tumor mutational burden (n.s. not statistically significant, two-sided Student's $t$ test). Source data are provided as a Source data file.

patients harboring mutations in these genes were more likely to present with bone metastasis at diagnosis compared to patients without MAP kinase pathway 5′-UTR mutations ($p < 0.05$) (Fig. 7d). Moreover, this difference was independent of 5′-UTR tumor mutational burden (Fig. 7e). As such, these findings demonstrate the potential clinical association of 5′-UTR alterations in mCRPC patients and highlight the importance of 5′-UTR mutations in oncogenic pathway activation and cancer progression.

## Discussion

Here, we describe the functional landscape of somatic 5′-UTR mutations at the transcript and translation levels in prostate cancer. We observed that 5′-UTR mutations affect a variety of cancer-associated pathways, some specific to localized disease, while others to metastatic disease. Moreover, these genetic variants are enriched in *cis*-regulatory elements encoded within specific 5′-UTRs, providing a mechanistic rationale for their existence. Within tumor specimens derived from patients, we found that somatic 5′-UTR mutations correlate with changes in transcript levels and translation rates of oncogenic gene targets independent of gene dosage. Moreover, we observed that 5′-UTR mutations within MAP kinase signaling pathway components are associated with pathway activation, response to chemotherapy, and early onset of lethal metastases. These findings implicate somatic alterations to leader sequences as a mechanism for deregulating the flow of genetic information, thereby enabling oncogenic levels of gene expression. While 5′-UTR mutations have been identified in a number of cancers, a major question in the field is what is the functional relevance of mutations within this noncoding space and how do they alter gene expression[6]? This is a critical question because recent studies have shown that the aggregate sum of putative passenger mutations, many of which lie in the 5′-UTR, have clinical consequences[11]. Although these findings point to an important association of these variants with a disease, functionally testing all full-length 5′-UTRs and alterations to determine their biological implications is undoubtedly needed. However, this has not been accomplished to date given the inherent limitations of traditional MPRAs as well as the need to quantify changes at both the transcript and translation levels.

In order to fill this experimental and conceptual gap, we conducted a functional genomic analysis of patient-based somatic 5′-UTR mutations across the spectrum of human prostate cancer. This was enabled by the development of PLUMAGE, a long- and short-read sequencing platform that assays full-length 5′-UTRs in a multiplex manner at both the mRNA transcript and translation levels simultaneously. Using this technology on mutations identified in prostate cancer patients, we demonstrate that 35% of mutations within the 5′-UTR can increase or decrease transcript levels or TE. Furthermore, we found through mechanistic studies that mutations within leader sequences can re-code their regional nucleotide context to promote oncogenic gene expression. For example, a simple C -> T mutation at chr15: 49715462 of the *FGF7* 5′-UTR was sufficient to increase transcript levels. This increase was mediated through the creation of an E-box motif, which we demonstrated enables MYC:MAX heterodimer binding. Importantly, we also showed that the results of our MPRA are congruent with endogenous gene expression changes using CRISPR base editing of a C -> T mutation in the 5′-UTR of *CKS2*. This mutation created a uAUG that increased translation of the mRNA in its endogenous context. Of note, while many 5′-UTR mutations we studied ablate or create new *cis*-elements, we also found alterations that were not associated with any known motifs and yet still cause changes in transcript abundance or TE. In this

context, it is interesting to speculate that these point mutations may instead affect local mRNA structure or epitranscriptomic marks, which can have profound changes in RNA metabolism and ribosome loading[6]. Future studies are needed to disentangle the effects of somatic 5′-UTR mutations on regional mRNA structure and their ability to be chemically modified.

Lastly, our work provides a resource and technology for multilayer functional genomic studies of genetic diseases. The versatility of the PLUMAGE methodology allows for customization to study cell-type-specific regulation of noncoding elements through lentiviral transduction. The assay can also be adapted to interrogate diverse variants in a variety of genomic regions, such as functionally characterizing all polymorphisms or variants of unknown significance in both the coding and noncoding genomic space. Thus, as a technological resource, PLUMAGE is poised to unlock previously untapped frontiers of human genetics.

## Methods

**Patient enrollment and tissue acquisition.** The Institutional Review Board of the University of Washington and the Fred Hutchinson Cancer Research Center approved all procedures involving human subjects. Tissue samples were obtained from male patients enrolled in the Prostate Cancer Donor Program at the University of Washington, who died of mCRPC. All patients in the study signed written informed consent for a rapid autopsy performed within 6 h of death. All visceral metastases were assessed at a gross level and were flash frozen and embedded in Optimal Cutting Temperature compound (Tissue-Tek)[22]. Eighty metastatic tumor samples and their corresponding matched normal tissue were obtained from individual patients. Normal prostate tissue of high glandularity was also obtained from five individuals (see Supplementary Fig. 2).

**PDX tissue.** The five LuCaP series of prostate cancer xenografts used in this study (LuCaPs 78, 81, 92, 145.2, and 147) were obtained from the University of Washington Prostate Cancer Biorepository and generated from advanced prostate cancer patients[23]. All relevant ethical regulations for animal testing and research were met. Mice containing xenograft tissue were maintained under specific pathogen-free conditions, and all experiments conformed to the guidelines as approved by the Institutional Animal Care and Use Committee of the University of Washington.

**Cell lines.** Human embryonic kidney 293T (HEK 293T) cells obtained from ATCC (CRL-3216) were cultured in Dulbecco's modified Eagle's medium (Gibco) supplemented with 10% fetal bovine serum (FBS) and 1% penicillin and streptomycin. The human prostatic carcinoma cell line PC3 obtained from ATCC (CRL-1435) was cultured in RPMI-1640 medium (Gibco) supplemented with 10% FBS and 1% penicillin and streptomycin. HEK 293T cells were chosen for their ease in transfection and robust expression of transfected plasmids. These cells are also commonly used in expressing high-complexity libraries and MPRAs. PC3 cells are a commonly used prostate cancer cell line, and were chosen because they are androgen-independent, show aggressive behavior, and are used to represent castration-resistant tumors. Both cell lines were grown at 37 °C in a humidified atmosphere containing 5% $CO_2$. A measure of 0.05% Trypsin-EDTA solution (Gibco) was used to detach cells from culture dishes. The cell cultures for HEK 293T and PC3 both tested negative for the presence of mycoplasma and were authenticated by short tandem repeat profiling and matched to STR profiles from the ATCC database for human cell lines.

**Genomic UTR sequencing.** Genomic DNA from frozen tissue was extracted using the Qiagen Gentra Puregene Tissue Kit (Qiagen). Sequencing libraries were prepped with the KAPA HyperPrep Kit (Roche) using 1 μg of DNA. DNA was sheared using a Covaris LE220 ultrasonicator targeting 200 bp, and sequencing adaptors added by ligation. Individually barcoded libraries were pooled 4-plex before capture. Libraries were hybridized to SeqCap EZ Choice probes of the 50 Mb Human UTR Design (Roche), and sequenced on a HiSeq 2500 (Illumina) using a PE100 in high-output mode. Image analysis and base calling were performed using Illumina's Real Time Analysis v1.18.66.3 software, followed by demultiplexing of indexed reads and generation of FASTQ files, using Illumina's bcl2fastq Conversion software v1.8.4.

**Ribosome profiling.** Flash-frozen human tumors dissected from each LuCaP PDX were manually pulverized under liquid nitrogen and lysed in 1 mL mammalian lysis buffer according to the TruSeq Ribo Profile (Mammalian) protocol (Illumina). Normal human prostate tissues from high glandular areas were obtained in the form of frozen shavings (200 mg) and lysed in the lysis buffer. To impede post-lysis translation, the lysis buffer was supplemented with cycloheximide (Sigma) dissolved in EtOH at a final concentration of 0.1 mg/mL. For complete tissue lysis, the

samples were further mechanically dissociated using a gentleMACS™ Dissociator (Miltenyi Biotec). Lysates were centrifuged, and the supernatants were used to isolate both total RNA and ribosome-bound fractions using the TruSeq Ribo Profile (Mammalian) Kit (Illumina). Ribosomal RNA was removed using the RiboZero Gold Magnetic Kit (Epicentre) before polyacrylamide gel electrophoresis (PAGE) purification. Ribosome footprints were generated by treating a portion of the lysate with 0.5 μL of TruSeq Ribo Profile nuclease per sample for 45 min at room temperature. The resulting monosomes were purified using Sephacryl S400 columns (GE Healthcare), from which ribosome-protected mRNA fragments were isolated and used to prepare ribosome footprint libraries. All libraries were quantified using the Qubit 2.0 fluorometer (Invitrogen), while the quality and average fragment sizes were estimated using a Bioanalyzer (High Sensitivity assay, Agilent). Barcodes were used to perform multiplex sequencing and create sequencing pools containing multiple samples with equal amounts of both total mRNA and ribosome footprints. The pools were sequenced on the HiSeq 2500 platform using SR50 sequencing chemistry.

**Luciferase assays**. To generate constructs for use in the luciferase reporter gene assay, primers containing NcoI and HindIII restriction enzyme sites were used to PCR amplify both the WT and mutant 5′-UTRs from complementary DNA (cDNA) generated from the PDXs, using the Phusion HiFi Master Mix (Thermo Fisher). These PCR products were purified by gel excision, digested with the NcoI and HindIII restriction enzymes (NEB), and cloned into the linearized pGL3-promoter-luciferase vector (Promega) using Quick Ligase (NEB) according to the manufacturer's protocol. The ligated product was transformed into chemically competent Escherichia coli, and plated onto LB agar plates containing ampicillin. Single bacteria colonies were inoculated into LB and grown overnight at 37 °C. Plasmid DNA was extracted from the bacteria cultures using the QIAprep Mini Kit (Qiagen), and Sanger sequenced to verify the cloned product. The successfully cloned plasmids containing the WT and mutant 5′-UTR sequences of interest were transfected into cell lines using Lipofectamine 3000 (Invitrogen) according to the manufacturer's protocol. Firefly luciferase activity was measured 24 h after transfection using the Dual-Glo Luciferase assay system (Promega) according to the manufacturer's instructions. Luminescence was measured on a BioTek Synergy HT (BioTek), and data were collected via the Gen5 2.01.14 software. Relative luminescence units (RLUs) from the luciferase assays were normalized against the amount of luciferase transcript by qPCR, as a quantitative readout of TE. Box plots show lines at the median, 25th, and 75th percentiles. Error bars reflect the minimum and maximum values.

**Quantitative PCR**. To validate changes in transcript levels brought about by 5′-UTR mutations, RNA and DNA were extracted from PC3 cells transfected with individual FOS, FGF7, and QARS WT and mutant plasmids using the AllPrep DNA/RNA Mini Kit (Qiagen). cDNA synthesis was performed on 1 μg of RNA using the SuperScript First-Strand Synthesis System (Invitrogen) and an RT primer (Supplementary Data 8). qPCR was performed on the DNA and cDNA using SsoAdvanced Universal SYBR Green Supermix (Bio-Rad) in triplicates, with primers against luciferase (Supplementary Data 8). To validate changes in mRNA translation, RNA and luciferase activity were collected from PC3 cells transfected with individual NUMA1, AKT3, and QARS WT and mutant plasmids. Total mRNA was extracted using the Quick-RNA Miniprep Plus Kit (Zymo Research), and cDNA synthesis and qPCR were performed as described. For the CKS2 experiment RNA was extracted from ~500,000 cells per 293T CKS2 WT or mutant cell line using the RNeasy Plus Kit (Qiagen) following the manufacturer's protocol. cDNA was synthesized using 500 ng RNA and iScript RT Supermix (Bio-Rad) or iScript NRT Supermix for negative controls. qPCR was performed using SsoAdvanced Universal SYBR Green Supermix (Bio-Rad) on 1 μL of each cDNA, no reverse transcription, and no-template control sample in triplicate using primers specific to CKS2 (primer sequences in Supplementary Data 8) and β-actin (primer sequences in Supplementary Data 8) as a housekeeping control.

**Construction of master pGL3 reporter backbone for PLUMAGE**. The pGL3-promoter-luciferase plasmid (Promega) was linearized using the XbaI restriction enzyme (NEB). A 202-bp double-stranded DNA fragment (IDT) containing an EcoRI restriction enzyme site followed by a 36-bp spacer sequence was cloned into the pGL3-promoter vector by Gibson assembly using the Gibson Assembly Master Mix (NEB) (sequence of 202-bp double-stranded DNA fragment:

AAGTACCGAAAGGTCTTACCGGAAAACTCGACGCAAGAAAAATCAGA GAGATCCTCATAAAGGCCAAGAAGGGCGGAAAGATCGCCGTGTAATtcta-gagaattctcatgtaattagttatgtcacgcagatcggaagagcGTCGGGGCGGCCGGCCGCTTCG AGCAGACATGATAAGATACATTGATGAGTTTGGACAAAC).

Successfully assembled plasmids were verified by Sanger sequencing. This master luciferase reporter backbone was then digested with both HindIII and EcoRI restriction enzymes (NEB) according to the manufacturer's instructions, and the larger fragment was gel excised, purified, and used as the backbone for cloning the PLUMAGE library.

**Generation of randomer barcoded master pGL3 reporter backbone**. Barcoded DNA fragments containing the luciferase gene were generated by PCR, using the

pGL3-promoter master reporter described above containing EcoRI and spacer sequences as a PCR template. An 80-bp oligonucleotide encompassing a semi-random 30-bp barcode sequence (15 repeats of A/T (W)–G/C (S)) was synthesized by IDT, and used as a reverse primer in the PCR reaction, along with a universal forward primer with sequences corresponding to the beginning of the luciferase gene. The PCR reaction was performed for 15 cycles, in 96-well plates, using the Phusion high-fidelity polymerase with HF buffer (Thermo Fisher). Following the PCR reaction, 1 μL of DpnI (NEB) was added to each well, along with Cutsmart buffer (NEB), and incubated at 37 °C for 45 min to digest the PCR template. A 96-well format DNA Cleanup and Concentrator Kit (Zymo Research) was used to purify the PCR reaction in each well according to the manufacturer's instructions. Each reaction was eluted in 21 μL of elution buffer. A total of ten 96-well plates of barcoded luciferase PCR products were generated.

**Full-length 5′-UTR DNA sequences**. A total of 914 full-length WT and mutant 5′-UTR sequences from 329 genes mutated in two or more patients or comprising oncogenic lesions were synthesized as double-stranded DNA fragments (IDT and SGI-DNA) (Supplementary Data 6a). Given the variability of transcription start sites (TSSs), putative TSSs of all 5′-UTRs assayed were confirmed by comparing the reference TSS (Refseq) with cumulative 5′-UTR reads of each gene across two independent prostate cancer RNAseq datasets. Each fragment was flanked with 36 bp of homology sequences for Gibson assembly. The homology sequence GAG-GAGGCTTTTTTGGAGGCCTAGGCTTTTGCAAAA was added to the 5′-end of each 5′-UTR sequence, while the other homology sequence CATGGAAG ACGCCAAAAACATAAAGAAAGGCCCGGC was added to the 3′ end of each 5′-UTR sequence. Sixty-nine out of 329 genes (20%) required a small modification to allow for synthesis. These small modifications involve the removal of repeat sequences and were completed for matched WT and mutant pairs.

**Cloning of PLUMAGE libraries**. Full-length 5′-UTR sequences and barcoded luciferase PCR products were cloned into the pGL3-promoter master reporter backbone using the Gibson Assembly HiFi 1-Step Kit (SGI-DNA). Each cloning reaction was carried out in each well in a 96-well plate, and consisted of 1 μL of barcoded PCR product, 1 μL of linearized master reporter backbone, 3 μL of 5′-UTR DNA fragment, and 5 μL of Gibson Assembly 1-Step Master Mix. For 5′-UTR sequences >1000 bp in length, 2 μL of DNA fragment and 2 μL of barcoded PCR product were used. The reaction was incubated at 50 °C for 1 h, after which 1.5 μL was transformed into 20 μL of 5-alpha chemically competent E. coli in 96-well plates (NEB) and transformed according to the manufacturer's protocol. One hundred and eighty microliters of room temperature SOC was added to each well and incubated at 37 °C for 90 min. The SOC transformants in each well were pooled from each 96-well plate, and 2 mL was plated onto a 500 cm2 LB agar plate containing ampicillin at a final concentration of 100 μg/mL. Three agar plates were used per 96-well plate to generate sufficient numbers of colonies to adequately represent each 96-well plate. To constrain the library size, ~300 bacteria colonies per well (or ~30,000 colonies per 96-well plate) were collected. Plasmid DNA was subsequently extracted using the Endotoxin-free Maxiprep Kit (Qiagen). The plasmid DNA concentration from each maxiprep was measured using the Qubit dsDNA HS assay (Thermo Fisher) and pooled in equimolar amounts to form a plasmid DNA library that consists of ~300,000 unique barcodes.

**SMRT sequencing of plasmid DNA library**. To verify the identity of each WT and mutant 5′-UTR, and to simultaneously associate it with unique 30-bp barcode sequences, the pooled plasmid DNA library was sequenced using long-read PacBio Sequel v3.0 sequencing chemistry (Pacific Biosciences). The plasmid DNA library was first linearized using the SalI restriction enzyme (NEB), which resides downstream of the 30-bp barcode. Since certain 5′-UTRs also harbor the SalI recognition sequence (GTCGAC), and will be truncated, given the restriction enzyme sequence can be found in genomic sequences, these were re-transformed into bacteria, harvested in a separate pool with ~300 bacterial colonies per transformation, DNA purified, and linearized with the BamHI restriction enzyme (NEB). Linearized plasmids from both pools ranging from 5000 to 7500 bp were size selected and eluted using the BluePippin system (Sage Science). DNA quantity of the eluates was measured for each pool (SalI and BamHI-generated pools) using an Agilent 4200 TapeStation, and 500 ng from each pool was used to prepare a SMRTbell library. Prior to ligation of the hairpin adapters that bind the sequencing primer and DNA polymerase, amplicons underwent damage and end repair to create double-stranded amplicon fragments with blunt ends. The resulting SMRTbell libraries were purified with PacBio AMPure PB beads, combined with a sequencing primer and polymerase, and loaded onto the SMRT cell. The SalI-generated pool was sequenced over three SMRT cells, while the BamHI-generated pool was sequenced over one SMRT cell.

**Construction of small proof-of-concept PLUMAGE DNA library**. This small library was constructed using a different cloning strategy, by utilizing a fixed number of known 8-bp barcode sequences. Luciferase plasmids containing full-length unmutated and mutated 5′-UTR sequences of ADAM32, COMT, and ZCCHC7 were linearized, and the 8-bp barcode was cloned at the end of the luciferase coding sequence by PCR. Each barcode was cloned in a separate cloning

reaction, transformed into chemically competent *E. coli*, and sequenced to determine successful assembly. Each plasmid with its unique 8-bp barcode was pooled in equimolar amount and transfected into PC3 cells. Long- and short-read sequencing were performed as described above. Box plots show lines at the median, 25th, and 75th percentiles. Error bars reflect the minimum and maximum values.

**Transfection of DNA library into 293T and PC3 cell lines**. A total of $2.6 \times 10^6$ 293T cells were plated onto a 15 cm dish, incubated overnight, and transfected with 16 µg of plasmid DNA library using Lipofectamine 3000 reagent (Invitrogen) according to the manufacturer's protocol. At 24 h after transfection, cells were washed with phosphate-buffered saline (PBS), harvested with 0.05% Trypsin-EDTA (Gibco), and centrifuged at $300 \times g$ for 5 min into a cell pellet. For the PC3 cell line, $3 \times 10^6$ cells were plated onto a 15 cm dish and transfected with 16 µg of plasmid DNA library using Lipofectamine 3000 reagent (Invitrogen) according to the manufacturer's protocol. At 24 h after transfection, cells were washed with PBS, harvested with 0.05% Trypsin-EDTA (Gibco), and centrifuged at $300 \times g$ for 5 min into a cell pellet. A total of $2.6 \times 10^6$–$3 \times 10^6$ cells/plate were chosen to enable over 900× coverage of the plasmid library per replicate (assuming 100 plasmids/cell, >25% transfection efficiency, and 212,325 unique constructs within the library). In both cell lines, cell pellets collected from each 15 cm dish were resuspended in 1 mL of cold PBS (Gibco) + 100 µg cycloheximide (Sigma) and incubated on ice for 10 min. The cells were centrifuged into a cell pellet and lysed in 220 µL of lysis buffer (Tris-HCl, NaCl, MgCl$_2$, 10% NP-40, Triton X-100, SUPERase In RNase Inhibitor, cycloheximide, dithiothreitol, diethyl pyrocarbonate water) for 45 min on ice, and vortexed every 10 min. For each cell line, lysates from three 15 cm dishes were pooled together to form one biological replicate. A total of three biological replicates were performed for each cell line. From each replicate, 60 µL of lysate was collected for DNA extraction using the QIAprep Spin Miniprep Kit (Qiagen). To collect total mRNA, 800 µL of Trizol (Life Technologies) was added to 150 µL of lysate and stored at −80 °C.

**Polysome profiling**. The remaining lysates from each biological replicate were centrifuged at $13,000 \times g$ for 5 min at 4 °C to pellet cell debris, and the supernatants were transferred into fresh tubes. A measure of 350 µL of the supernatant was layered onto 10–50% (w/v) sucrose gradients for ribosome fractionation. The gradients were centrifuged at $170,000 \times g$ for 2.5 h at 4 °C in a Beckman SW41Ti rotor and fractionated by upward displacement into collection tubes through a Bio-Rad EM-1 UV monitor (Bio-Rad) for continuous measurement of the absorbance at 254 nm using a Biocomp Gradient Station (Biocomp). 80S and polysome samples were collected and subsequently processed for sequencing. In particular, polysome fractions (three or more ribosomes) were pooled; RNA extracted from this pool was compared to total mRNA to determine TE. In addition, the pool of polysome fractions was also compared to 80S-bound mRNA as an alternate measure of translation.

**RNA extraction and cDNA synthesis**. Total, 80S-associated, and polysome-bound RNA were extracted using the Direct-zol RNA Miniprep Plus Kit (Zymo Research) following the manufacturer's protocol including the on-column DNase digestion. For polysome, RNA samples after the disome were pooled before RNA extraction. To ensure that there was no plasmid carryover and that we were truly detecting mRNA expression in our assay, an additional DNase treatment was performed on 2 µg of extracted RNA using 3 µL of DNase1 Amplification Grade (Invitrogen) in a total reaction volume of 20 µL, at room temperature for 30 min. The reaction was terminated by the addition of 2 µL of 25 mM EDTA with a 10-min incubation at 65 °C. Of this DNase-treated RNA, 8 µL was used in a cDNA synthesis reaction using the SuperScript III First-Strand Synthesis System (Invitrogen) with a primer specific to the 3′ end of the 30-bp barcode (sequence in Supplementary Data 8). Negative control reactions without the SuperScript III reverse transcriptase enzyme were also performed on all the RNA samples and confirmed to be negative. Reactions were incubated according to the manufacturer's instructions.

**Illumina sequencing library preparation**. Sequencing libraries were generated by performing first- and second-round PCRs on each DNA, and cDNA generated from the total, 80S-associated, and polysome-bound RNA samples. First-round PCR primers contain target-specific sequences flanking the 30-bp randomer barcode and Illumina adaptor sequences, producing a product of 215 bp (for sequences see Supplementary Data 8). The first-round PCR reaction was performed using 2× Phusion Flash Master Mix (Thermo Fisher) in a 50 µL reaction. The PCR reaction consisted of 5 µL of DNA or cDNA template, 2 µL of forward primer (10 µM), 2 µL of reverse primer (10 µM), and 25 µL of Phusion Flash Master Mix. Thermal cycling conditions were at 95 °C for 3 min, 20 cycles of 98 °C for 10 s, 60 °C for 30 s, 72 °C for 30 s, followed by 72 °C at 5 min. A small portion (3 µL) of the PCR products and negative controls were run on a 1.5% agarose gel for visual inspection. The first-round PCR products were purified using a 0.8× AMPure XP (Beckman Coulter) cleanup following the manufacturer's protocol with 80% ethanol. Following cleanup, 4 µL of the purified first-round PCR product was used as a template in the second-round PCR reaction. The forward primer contained the Illumina adaptor sequence, as well as the flow cell attachment sequence, and the

reverse primer contained an 8-bp index between the adaptor sequence and flow cell attachment sequence (Supplementary Data 8). The second-round PCR reaction was carried out in a 50 µL reaction similarly, using Phusion Flash Master Mix (Thermo Fisher), with 5 µL of each forward and reverse primer (0.5 µM). Thermal cycling conditions were at 95 °C for 3 min, eight cycles of 95 °C for 30 s, 55 °C for 30 s, 72 °C for 30 s, followed by 72 °C at 5 min. PCR products were purified using a 0.8× AMPure XP (Beckman Coulter) cleanup following the manufacturer's protocol with 80% ethanol. A sample (3 µL) of the purified PCR products was run on a 1.5% agarose gel for visual inspection. Each sample was quantified by qPCR using the KAPA Library Universal Quantification Kit (KAPA Biosystems) according to the manufacturer's instructions and pooled in equimolar amounts for multiplex sequencing. The final pool was denatured and diluted to a loading concentration of 7.5 pM as per the Illumina protocol. The PhiX control library (Illumina) was spiked in at 20% to add diversity for improved cluster imaging.

The libraries were sequenced employing a paired-end, 100 base read length (PE100) sequencing strategy on a HiSeq 2500 (Illumina). Image analysis and base calling were performed using the Illumina's Real Time Analysis v1.18.66.3 software, followed by demultiplexing of indexed reads and generation of FASTQ files, using Illumina's bcl2fastq Conversion Software v1.8.4.

**Electrophoretic mobility shift assay**. MYC and MAX were translated individually or together in vitro using the TnT SP6 coupled wheat germ extract system (Promega), according to the manufacturer's protocol. Plasmids used for MYC and MAX were pCS2-FLAG-hMYC and pRK7-HA-hMAX, respectively, and were generously provided by the Eisenman Lab (Fred Hutchinson Cancer Research Center). The protein concentrations of the in vitro translated products were determined using the Pierce BCA Protein Assay Kit (Thermo Fisher Scientific). Binding reactions were carried out using Odyssey EMSA Buffer Kit (LI-COR), where 90–100 µg of the translated proteins were incubated with 7.5 nM IRDdye 700-labeled FGF7 WT or mutant DNA probes (IDT) in the presence or absence of their respective unlabeled competitor oligos (IDT), according to the manufacturer's protocol. To separate the DNA–protein complex, the binding reactions were subjected to electrophoresis on a 6% DNA retardation gel (Thermo Fisher Scientific), which was then scanned using the Odyssey Infrared Imaging System (LI-COR) to detect the fluorescence signal. The assay was performed three times and showed similar results.

**Actinomycin D RNA degradation study**. Ten micromoles of actinomycin D (Sigma, prepared in dimethyl sulfoxide (DMSO)), or an equivalent volume of DMSO (Gibco) as a control, was added to PC3 cells in culture 48 h after transfection with WT or mutant plasmids. Cells were harvested prior to actinomycin D treatment, and again after 1 h of treatment. RNA was extracted for cDNA synthesis and subsequent qPCR amplification and quantitation of luciferase mRNA expression.

**CRISPR base editing**. Plasmid to express *CKS2*-targeting sgRNA was cloned using the Q5 Site-Directed Mutagenesis Kit (NEB) according to the manufacturer's instructions. The pFYF1320 sgRNA expression plasmid was used as a template for Q5 mutagenesis PCR (for primer sequences see Supplementary Data 8) to replace the existing sgRNA sequence with the CKS2-targeting sgRNA sequence (CTGGACGTGGTTTTGTCTGC).

293T cells were plated in six-well plates at 375,000 cells/well, incubated at 37 °C overnight, and transfected with 1125 ng evoAPOBEC1-BE4max-NG (Addgene: 125616), 375 ng *CKS2* sgRNA expression plasmid, and 30 ng pMaxGFP using Fugene HD (Promega) according to the manufacturer's protocol. At 72 h post transfection, cells were washed with PBS, harvested with 0.05% Trypsin-EDTA (Gibco), and centrifuged at $400 \times g$ for 5 min. This cell pellet was resuspended in PBS and sorted using flow cytometry for live, singlet, GFP+ cells on a Sony SH800 sorter. GFP+ cells were plated using limiting dilution in 10 cm plates to grow out single-cell clones.

After clones had grown sufficiently (~3 weeks), DNA was extracted using Zymo's MicroPrep Quick-DNA Kit, the *CKS2* locus PCR amplified using the Phusion High Fidelity Master Mix (Thermo Fisher) in a 25 µL reaction and primers (for primer sequences see Supplementary Data 8) according to the manufacturer's protocol. PCR products were then Sanger sequenced to determine if the intended *CKS2* mutation (chr9: 91926143 C -> T) had been introduced. Six individual clonal cell lines were chosen for further testing: three mutant clones each mutated at one of two *CKS2* alleles, and three WT clones that were not mutated.

**shRNA knockdown**. An shRNA construct targeting *CKS2* (hairpin sequence: TG CTGTTGACAGTGAGCGAACAGCAACAGAGCTCAGTTAATAGTGAAGCC ACAGATGTATTAACTGAGCTCTGTTGCTGTGTGCCTACTGCCTCGGA) in the pGIPZ backbone was obtained as a gift from the Paddison Lab (Fred Hutchinson Cancer Research Center). The shCKS2 construct was transfected into the *CKS2* mutant 2 clonal cell line created by CRISPR base editing due to its high endogenous expression of CKS2. Transfection was performed by plating 375,000 cells per well in six-well plates, incubating overnight at 37 °C, and next day adding 1.5 µg of plasmid DNA with 4.5 µL Fugene HD (Promega) according to the

manufacturer's instructions. At 24 h post transfection of shCKS2, cells were harvested and lysed for Western blotting.

**Western blotting.** A total of $1 \times 10^6$ cells were collected from each *CKS2* WT and mutant 293T cell line and lysed in RIPA lysis buffer (Thermo Scientific) supplemented with 10% Complete Mini Protease Inhibitor (Sigma) and 10% PhosSTOP Phosphatase Inhibitor (Roche). After incubating on ice for 30 min, lysates were centrifuged at $13,000 \times g$ for 10 min at 4 °C. The supernatant was collected and protein concentration was measured using a Bradford assay (Bio-Rad). Twenty-five to 50 μg of extract per cell line was separated by sodium dodecyl sulfate-PAGE and transferred onto PVDF membranes for immunoblot analysis. Primary antibodies used were CKS2 (Abcam, catalog #: ab155078; dilution: 1:1000) and β-actin (Sigma-Aldrich; catalog #: A5316; dilution: 1:1000).

**Bioinformatics analysis**

*Obtaining publicly available 5′-UTR sequencing data used for analysis.* BAM files 101 tumor/matched normal CRPC metastases patients were obtained from Quigley et al.[26] and bedtools "bamtofastq" (https://bedtools.readthedocs.io/en/latest/content/tools/bamtofastq.html) was used to extract raw sequencing data from BAM files.

*Genomic UTR sequencing alignment and quality control.* Raw sequencing reads produced by Illumina's bcl2fastq 1.8.4 software were processed to exclude read pairs failing default (PF filtering) quality checks. FastQC (v0.11.9) (https://www.bioinformatics.babraham.ac.uk/projects/fastqc/) was used to evaluate raw sequencing reads. All reads were aligned to hg19 using Bowtie (v1.0.0)[48], only reads aligning to standard chromosomes (chr1:22, X, Y, M) were retained for further analysis. The reads with low quality were filtered and duplicates were marked using Picard (v2.21.6). 5′-UTR and CDS coverage was calculated using GATK "DepthOfCoverage" (https://gatk.broadinstitute.org/hc/en-us/articles/360041851491-DepthOfCoverage). ContEst (v3.6)[49] was used to estimate the level of cross-individual contamination in matched normal-tumor pairs from all mCRPC patients and PDX samples that were sequenced.

*Mouse subtraction for human PDX 5′-UTR mutation calling.* Short reads from the LuCaP PDX specimens were aligned to both human reference genome hg19 and mouse reference genome mm9 separately using TopHat[50] (v2.0.14). An in-house developed software was applied to retain the reads with higher fidelity to hg19 for further downstream analysis.

*Somatic mutation analysis.* MuTect (v1)[51] and Strelka (v1)[52] were used to identify somatic single-nucleotide variants within the 5′-UTR and CDS for each tumor and matched normal pair. Two different bed files were used in two separate runs for obtaining 5′-UTR mutations and CDS mutations, 120501_hg19_RefSeq_Ensembl_UTRs_EZ_HX1_target_regions.bed and 120430_b37_ExomeV3_UTR_EZ_HX1_Union.bed, respectively. For the University of Washington rapid autopsy specimens, LuCaP, TCGA, and ICGC samples, the following cutoff was applied to derive a final list of mutations: total number of reads in tumor >14, and >8 in matched normal sample, number of reads aligning to alt allele in tumor ≥ 4 and tumor variant allele frequency (VAF) > 0.1. VAF refers to the fraction of sequencing reads overlapping a genomic coordinate that supports the non-reference (mutant/alternate) allele.

*Ribosome profiling data analysis.* Libraries were sequenced on Illumina HiSeq 2500 at the Genomics Shared Resource in the FHCRC. The raw sequence data were uncompressed followed by clipping the 3′ adaptor sequence (AGATCGGAAGAG CACACGTCT). Next, the trimmed sequence reads were aligned to human ribosomal RNA (rRNA) reference using Bowtie. The unaligned reads were collected, while the rRNA alignments were discarded to reduce rRNA contamination. TopHat (v2.0.14)[50] was used to align the non-rRNA sequencing reads to hg19 and subtraction of mouse sequences was performed using a custom script. Aligned reads were counted for gene associations against the UCSC genes database with HTSeq (0.11.0)[53]. Five LuCaP and five normal prostate tissue samples were sequenced twice. In each analysis, two replicates for each LuCaP were considered as the test group and five normal prostate tissue samples as the control group. Xtail (v1.1.15)[54] and DESeq2 (v1.26.0)[55] were both used to find translationally regulated genes individually for each LuCaP (FDR < 0.1 and fold change >1.5). Translation fold changes were highly correlated across both packages. Similarly, DESeq2 was used to find transcriptionally regulated genes individually for each LuCaP (FDR < 0.05 and fold change >2), which were excluded from the translationally regulated gene lists. R/Bioconductor package, riboseqR (v1.26.0) (http://bioconductor.org/packages/release/bioc/html/riboSeqR.html), was used to calculate triplet periodicity in all samples. GSEA was done using Broad's website for GSEA (http://www.gsea-msigdb.org/gsea/msigdb/index.jsp) using the Molecular Signatures Database (MSigDB).

*Extracting 5′-UTR sequences.* Using R/Bioconductor package GenomicFeatures[56] transcript ids, genomic coordinates and TSSs for 5′-UTR of each of the mutated genes were obtained from UCSC's Refseq Table. 5′-UTR sequences were retrieved

using R/Bioconductor packages "BSgenome.Hsapiens.UCSC.hg19". Coverage for 5′-UTR coordinates for each transcript was calculated using bedtools coverage (https://bedtools.readthedocs.io/en/latest/content/tools/coverage.html). Median coverage for 200 bp flanking putative TSSs was plotted using R.

*Cis-regulatory element mutational analysis.* Analysis of 5′-UTR mutations within *cis*-element regulatory regions was performed by examining if the observed mutations in our patient cohort disrupt DNA-binding element motifs, and other known elements, including RNA-binding protein-binding sites, upstream start codons (uAUGs), TOP-like/PRTEs, G-quadruplexes, and 5′-TOP. A custom set of Python scripts (available here: https://github.com/lukascorey/5-UTR-Mutation-Analysis) was written to determine whether the observed counts of 5′-UTR mutations were statistically enriched within these regulatory elements when compared to a random model preserving sequence-specific characteristics such as trinucleotide context. In this analysis, mutations that impacted pre-existing regulatory elements or introduced new elements were both considered. To generate the background distribution, permutations of all 5′-UTR mutation locations found within our dataset were performed ~10,000 times. The original mutational frequency of all specific transversions, transitions, and trinucleotide context (a total of 288 possible mutations are possible under this scheme—64 possible codons plus 32 additional with no nucleotide in exclusively the first or third position, each with three possible mutations to the middle base) were taken into account. The number of mutations in these permutations that affected each type of motif in each database or specific element was counted. The total number of observed mutations impacting each regulatory element type was compared to the background distribution of the permutation data and the *p* value was computed. We selected elements with $p < 0.025$ as being statistically significant.

*DNA-binding elements.* Position-weighted matrices of DNA-binding elements were retrieved from the HOMER database. Position frequency of all known motifs in these databases was converted to position-weighted matrices using the standard conversion ($\log_2$(frequency/0.25)). A total of 332 motifs were obtained from HOMER. All analyses with these motifs used a cutoff at 90% of the maximum score. Both the forward and reverse strands were scanned.

*RNA-binding proteins.* Position-weighted matrices of RNA-binding protein-binding sites were retrieved from the Hughes lab dataset[32]. Similarly, the position frequency of all known human motifs in these databases was converted to position-weighted matrices using the standard conversion ($\log_2$(frequency/0.25)). The analysis included 102 motifs from the Hughes database, with a 90% cutoff.

*Translational regulatory elements.* To assess the functional impact on uORFs, predicted functional uORFs were used[57]. Observed 5′-UTR mutations from our dataset or the simulated permutations that landed within a start codon of one of these predicted reading frames were counted as mutating a uORF. The PRTE motif consists of an invariant uridine at position 6 flanked by pyrimidines and does not reside at position +1 of the 5′-UTR and is similar to the TOP-like sequence. The provided position-weighted matrix was used to identify PRTEs. 5′-TOP tracts were characterized as regions at the 5′ end of a 5′-UTR beginning with a cytosine and followed by no fewer than four pyrimidines. Mutations in the first ten base pairs of a UTR with a 5′-TOP were counted as mutating that 5′-TOP. G-quadruplexes, defined as regions with four groups of at least two adjacent guanines separated by loops of at least one nucleotide but no more than seven nucleotides, were also considered in this analysis. For all RNA-binding proteins and translational regulatory elements, the analysis was performed on the single-stranded mRNA plus strand.

*PLUMAGE long-read sequencing analysis.* Associations between inline 30-bp barcodes and specific 5′-UTR sequences were established by long-read sequencing on the PacBio Sequel system using four SMRT cells. An additional SMRT cell was dedicated to a smaller pool of 5′-UTR targets, containing *Sal*I restriction sites, that were expected to be truncated in the main pool.

Subreads for each sequenced molecule were combined to form high-quality circular consensus sequences (CCS) using PacBio's Circular Consensus Sequencing 2 algorithm with default parameters (PacBio SMRTLink 6.0.0.47841, minimum three passes, minimum predicted accuracy 0.9). Within each CCS sequence, we identified the 5′-UTR and associated 30-bp barcode by searching for flanking 20-bp sequences expected to be constant across all constructs. CCS sequences where these flanking sequences were not found, or where a barcode had not been inserted and the *Eco*RI target sequence GAATTC remained, were excluded from further consideration.

Where available, 5′-UTR sequences that share the same barcode were combined by multiple alignments with MUSCLE (v3.8.31), generating a single consensus sequence for each observed barcode. Consensus 5′-UTR sequences were annotated by exact matching to the PLUMAGE sequences submitted for synthesis. Exact matching is required because the majority of the mutants differ by only a single base from the WT. Consensus sequences that did not match exactly any PLUMAGE sequence were annotated to the nearest PLUMAGE gene by blastn search, allowing us to identify genes whose 5′-UTRs may be difficult to sequence due to composition, repeats, or length.

The above process generated 968,990 CCS2 sequences containing 330,199 distinct 30-bp barcodes. Of these, 212,325 were associated with an exact match to an expected PLUMAGE 5′-UTR sequence. On average, annotated 5′-UTR sequences are supported by 236 distinct 30-bp barcodes (median is 200). Of the remaining 117,874 barcodes that did not match an expected 5′-UTR, 50% were supported by a single CCS2 sequence only so that multiple independent CCS2 sequences were unavailable for multiple alignment and further refinement. All 5′-UTR sequences (mutated and unmutated) identified by long-read sequencing are documented in Supplementary Data 6a. All unique 30-bp barcodes associated with each correctly synthesized 5′-UTR sequences were identified and used in the short-read sequencing analysis.

*PLUMAGE short-read sequence analysis*. To quantify 5′-UTR sequences in DNA, total mRNA, and polysome-bound mRNA, each sample was sequenced in triplicate on an Illumina HiSeq 2500 (PE100). Sequencing targeted only the barcode region of each sample ensuring that the barcode was completely contained within, and at a fixed offset from the 3′ end of the second 100 nt read in each pair. Barcodes were extracted from this fixed position, subject to the constraint that a short sequence (4 nt) on both sides match the expected sequence as a check on improper barcode length or placement. Using this method, barcodes were extracted from 80% of the reads in each sample, and >96% of the extracted barcodes matched one previously cataloged by PacBio long-read sequencing. Between 6.4 and 16.5 million cataloged barcodes were assigned to each sample in this way. Extracted barcodes were tallied against corresponding 5′-UTRs using the barcode-to-variant mapping generated from PacBio long-read sequencing. To determine the robustness of the assay, for each cell line, the number of times each barcode was observed, and the total number of barcodes observed for each 5′-UTR in each sample were counted. In addition to tables of raw counts, we produced counts-per-million (CPM) scaled summaries wherein raw counts were divided by the total number of reads (in millions) matched to barcodes in each sample to account for variation in sequencing depth. For each barcode, raw read counts were normalized by counts per million (CPM) within each sample for each biological replicate. All barcodes in each sample used in the calculation of ratios had a minimum normalized read count of 0.5 CPM. To determine changes in transcription, the $\log_2$ (total mRNA/DNA) CPM was calculated for each barcode within each biological replicate, and all the barcodes for the mutant 5′-UTR were compared to the corresponding WT 5′-UTR. A two-sided Mann–Whitney $U$ test was performed for each mutant and WT 5′-UTR using the R function wilcox.test and $p$ values were adjusted for multiple comparisons using the FDR method. Significance was determined by using a cutoff of FDR < 0.1. To determine changes in mRNA TE or polysome to 80S ratio, the $\log_2$ (polysome/total mRNA or polysome/80S) CPM was calculated for each barcode, and differences in mutant vs WT 5′-UTRs were determined in a similar manner. Significance was also determined by using a cutoff of FDR < 0.1. To demonstrate reproducibility, scatter plots of normalized counts for each unique barcode were made comparing each sample for each biological replicate. The Pearson correlation was calculated for each comparison using R function cor(). Density plots were made to represent normalized counts per barcode per sample using the R package ggplot2 (v.2.2.1).

*Copy number analysis*. Sequenza[58] (v2.1.9999b) was used to estimate allele-specific copy number calls, tumor cellularity and tumor ploidy for each tumor, and its matched normal sample. Average depth ratio (tumor vs normal) and B allele frequency (the lesser of the two allelic fractions as measured at germline heterozygous positions) was used to estimate copy number while considering the overall tumor ploidy/cellularity, genomic segment-specific copy number, and minor allele copy number. Approximately 150 bp sequences flanking the 5′-UTR mutation were considered.

*GSEA analysis*. MSigDB (v1.7) was used to compute overlaps with Kyoto Encyclopedia of Genes and Genomes (KEGG) gene sets present in MSigDB database, and gene sets with FDR < 0.05 were considered significant. Fisher hypergeometric tests were implemented in R using function phyper() to see if genes in one set were over-represented, compared to other gene sets.

*MAP kinase network visualization*. MAP kinase signaling pathway (map0410) was downloaded from KEGG, and Cytoscape (v3.7.2, https://cytoscape.org/) was used to visualize the network where genes mutated in metastatic samples were colored in green and non-mutated genes were colored in gray.

*RNAseq data analysis*. Raw sequencing from Nyquist et al. Cell Reports 2020[59] was aligned to hg19 using TopHat (v2.0.14), and aligned reads were counted for gene associations using HTSeq against the UCSC genes database. Normalized RNAseq data from Nyquist et al. 2020[59] and mRNA samples from LuCaPs were used to conduct a GSVA analysis for all C2 canonical pathways (KEGG, BIOCARTA, and REACTOME) from MSigDb. A no-scale heatmap representing GSVA results for MAP kinase pathways was made using R package pheatmap (v1.0.12) (https://cran.r-project.org/web/packages/pheatmap/). With the same samples, GSVA analysis was also conducted using genes upregulated in various mouse prostate tumors from Wang et al. Cancer Research 2012[45] and represented as a color bar on the heatmap, as a MAP kinase pathway activity score.

*Statistical analysis*. All box plots and violin plots have the median as the center, and the first and third quartiles as the upper and lower edges of the box. All minimum and maximum data points are shown. Sample sizes, biological replicates, and $p$ values are indicated in relevant figures. All $p$ values were obtained from two-tailed Student's $t$ tests, except for PLUMAGE validation experiments, where the one-tailed $t$ test was used to assess known directionality. A two-sided Mann–Whitney $U$ test was performed for each mutant and WT 5′-UTR in PLUMAGE short-read sequencing data analysis. All $p$ values were adjusted for multiple comparisons using the FDR method. Pearson correlation was calculated for comparisons between replicates and cell lines. The Fisher hypergeometric test was used to determine statistical significance between different gene sets.

**Reporting summary**. Further information on research design is available in the Nature Research Reporting Summary linked to this article.

## Data availability

All data needed to evaluate the conclusions in the paper are described in the paper and/or the methods, and are available at the following repositories or databases. The Fraser et al.[21] publicly available data used in this study are available in the European Genome-Phenome Archive under accession code EGAD00001003139. The 20 tumor/matched normal localized prostate cancer patients (WGS) publicly available data used in this study are available through the Database of Genotypes and Phenotypes (dbGaP) under accession code phs000178.v11.p8. The Quigley et al.[26] publicly available data used in this study are available in dbGaP under accession code phs001648.v2.p1. The Nyquist et al.[59] publicly available data used in this study are available in the Gene Expression Omnibus (GEO) under accession code GSE147250. The DNA-binding elements are publicly available data used in this study and are available in the HOMER database (http://homer.ucsd.edu/homer/). The RNA-binding elements are publicly available data used in this study and are available from the Ray et al.[32] publication (http://hugheslab.ccbr.utoronto.ca/supplementary-data/RNAcompete_eukarya/) and the CISBP database (http://cisbp.ccbr.utoronto.ca/). The predicted functional uORFs are publicly available data used in this study are available from the McGillivray et al.[57] publication (https://academic.oup.com/nar/article/46/7/3326/4942470#supplementary-data). The MAP kinase signaling pathway publicly available data are available in KEGG under accession code map04010. The human mCRPC UTR sequencing data generated in this study have been deposited in dbGaP under accession code phs001825.v1.p1. The data are available under restricted access; access can be obtained by following the instructions on dbGaP. The RNAseq, ribosome profiling, Exome-seq data for PDXs, and PLUMAGE short-read sequencing data generated in this study have been deposited in GEO under SuperSeries GSE149489. The remaining data are available within the Article. Source data are provided with this paper.

## Code availability

Scripts used in this publication were developed in-house and are available at: https://github.com/sonali-bioc/Lim-5utr-Paper/releases/tag/v1.0.0 (https://doi.org/10.5281/zenodo.4773696); code for 5′-UTR *cis*-regulatory element mutation analysis can be found at: https://github.com/lukascorey/5-UTR-Mutation-Analysis/releases/tag/v1.0 (https://doi.org/10.5281/zenodo.4774281).

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

## Acknowledgements

We would like to thank the members of the Hsieh laboratory, P. Paddison, and R. Subramanium for helpful discussion on creating PLUMAGE. We thank S. Beronja and C. Bellodi for their critical reading of the manuscript. We also thank the patients and their families, E. Yu, El. Mostaghel, H. Cheng, B. Montgomery, M. Schweizer, P. Lange, D. Lin, M. Roudier, R. Vessella, C. Morrissey, and the rapid autopsy teams for their contributions to the University of Washington Medical Center Prostate Cancer Donor Rapid Autopsy Program and the development of the LuCaP PDX models. This work was supported by National Institute of Health grants R37 CA230617 (to A.C.H.), R01 DK119270 (to A.C.H.), R01 GM 135362 (to A.C.H.), P50CA097186, U54 CA224079, R01 CA234715, and CDMRP awards W81XWH-18-1-0406 and W81XWH-14-2-0183. This research was also supported by the Genomics and Bioinformatics Shared Resource of the Fred Hutch/University of Washington Cancer Consortium (P30 CA015704) and the Scientific Computing Infrastructure at Fred Hutch funded by ORIP grant S10 OD028685. Establishment and characterization of the LuCaP models was supported by the PNW Prostate Cancer SPORE and the IPRC. A.C.H. is a recipient of a Prostate Cancer Foundation Challenge Award (16CHAL03), a Burroughs Wellcome Fund, Career Award for Medicine Scientists (1012314.02), and grants from the Emerson Collective (691630) and the Robert J. Kleberg Jr. and Helen C. Kleberg Foundation. Y.L. was supported by the 2016 AACR-Bristol-Myer Squibb Oncology Fellowship and the Department of Defense CDMRP Prostate Cancer Postdoctoral Training Award. S.L.S. is supported by NIH NRSA grant F31 CA260920. X.W. is supported by R35 CA231989.

## Author contributions

Y.L. and A.C.H. conceived and designed all experiments. Y.L. performed all experiments and analyzed results. S.A., L.C., and M.F. conducted the computational analyses. L.C., S.L.S, C.L.W., and X.W. performed validation experiments. J.J.D. provided technical and computational guidance. I.M.C., E.C., L.D.T., and P.S.N. provided PDX, patient samples, and information. G.H. provided computational guidance. Y.L. and A.C.H. wrote the manuscript. All authors reviewed and edited the manuscript.

## Competing interests

The authors declare no competing interests.
