## [Peer Review File · Nature Communications]

REVIEWER COMMENTS

Reviewer #1, expert in 5' UTR mutations (Remarks to the Author):

In this study, Lim et al. developed a high-throughput method called PLUMAGE to investigate the effects of mutations in 5' UTR of mRNAs in prostate cancer patients on translation and transcription. By using PLUMAGE, they identified hundreds of mutations in 5' UTR that can significantly alter translation or transcription levels. Further computational analyses showed that these cancer-associated mutations are mainly located to the cis-elements. Besides, they reported that even a single mutation within cis-elements is sufficient to change translation or transcription levels. Although massively parallel reporter assays have been intensively used to investigate the relationship of sequence contexts to transcription or translation, PLUMAGE provides a new way to study translation and transcription simultaneously, by estimating the relative abundance of 3' end tethered unique sequences. The using of long-read and short-read sequencing technologies to couple the WT and mutated 5' UTR with the unique barcodes is clever and interesting. However, the current version is a bit premature, and some major concerns need to be addressed.

My major concern is they used the barcodes (8 nt or 30nt) in the 3' end of the reporter to estimate transcription and translation activity, by isolating total RNAs and ribosome associated RNAs. There are a few potential problems of this design.

1. The random barcodes in 3' UTR may alter translation efficiency as well as mRNA stability. The authors should show evidence that any translation changes associated with 5' UTR mutations they observed are not due to random barcodes in 3'UTR.
2. Change of RNA levels may stem from either transcriptional regulations or post-transcriptional regulations, e.g. RNA stability. Although they reported that many mutations can be located to cis-elements, suggesting that lots of mutations are likely to alter transcriptional regulations, mutations on RNA stability need to be excluded or normalized.
3. The authors used sucrose gradient to isolate ribosome associated mRNAs. I did not find in method sections which fraction they used, monosome or polysome? Or all the fractions? I wondered the sensitivity of this method to estimate translational changes. For example, a WT reporter mRNA is translated by monosome, whereas the mutation reporter mRNA is translated by multiple ribosomes, i.e. translation is increased by mutations. However, the two reporter mRNAs are classified into ribosome associated mRNAs, thus may not have significant difference in detected RNA levels. They may use a similar method (mean ribosome loading, MRL) in PMID: 31267113 to increase the sensitivity of reporter.

Below are my other concerns.

1. Figure 1g. The three mutations were tested to demonstrate results from PLUMAGE are consistent with the results from ribosome profiling (Figure 1d). It would be better if a transcriptome wide correlation between PLUMAGE and ribosome profiling is shown.
2. Figure 2e. Are the mutations that significantly alter translation efficiency enriched in the TOP mRNA?
3. Figure 3c and 3d. Although the correlation between cell lines remains high, the significantly reduced correlation indicates that the effect of some mutations could be cell line dependent.
4. Are the flanking sequences surrounded the mutations that significantly alter transcription or translation activity subjected to negative natural selection, compared with neutral mutations?

Reviewer #2, expert in prostate cancer genomics (Remarks to the Author):

SUMMARY:

In this study Lim et al. have examined mutations in the 5' UTR in 5 PDX models of prostate cancer and 229 patient samples from both localized and mCRPC. They characterized the gene families affected by these mutations and also their potential to disrupt known DNA and RNA binding motifs. They then examined the effect of these mutations on transcription and translation via a novel high-throughput technique they named PLUMAGE and validated some candidates orthogonally. Lastly, they described mutations in MAPK pathway genes which are only seen in mCRPC.

The work in this manuscript is well-presented and highly relevant. There is still major gaps in knowledge about how alterations in non-coding regions of the genome, including UTRs, affect gene expression. As such, their findings would be of interest to many researchers not just those in the prostate cancer field. Particularly novel is their new high-throughput method PLUMAGE which allows for the simultaneous measurement of the effect of mutations of expression AND translation. Aside from specific major and minor comments delineated below, the manuscript would be improved by focusing more on the positive results that came out of the PLUMAGE screen of mutations. Functional experiments manipulating these sites by creating isogenic cell lines would be the gold standard, and would allow for the determination of the effects in the native context of that transcript's stability. However, even in the absence of further functional data it would be satisfying to have more emphasis on the mutations that result in significant changes: what cancer-related pathways would they regulate and in what direction, is there anything that delineates them from the non-significant ones, is there a difference in localized vs CRPC, are they specifically associated with any clinical parameters? As written the authors answer many of these questions only for their list of overall mutations, despite the fact that their unique analysis with PLUMAGE is what truly differentiates their study.

MAJOR COMMENTS:

1) Figure 1C and 4C- In these two panels the authors are trying to associate changes in the gene expression or translation to the presence of the 5' UTR mutations. In order to do so they compare expression or TE in each line to 5 normal prostate samples. These comparisons are flawed in that it would appear that a large percentage of differences that are picked up as significant are actually just demonstrating PCa vs normal instead of differences due to any mutations. For example, as highlighted in Fig 1C, a mutation in ADAM32 in LuCaP147 is associated with decreased TE in that sample. However, in Supplementary Table 1C they report that TE is also decreased in 3 of the other 4 LuCaP PDX despite the fact that they do not have the 5'UTR mutation. These issues persist with other "significant" changes - in LuCaP78 3 genes are reported to have mutations that alter TE, but 2 of these genes show altered TE in other PDX that don't have the mutation. In LuCaP145 all of the TE changes linked to mutations are seen in other PDX. The authors should address and provide any statistical support that would confirm mutation association with TE.

2) Figure 5- After having run PLUMAGE to find which of the 5'UTR mutations result in functional changes in expression or translation the authors now go back to their original list of all the mutations identified. This is especially confusing given that the PLUMAGE analysis showed that only about a third of the mutations caused any significant change. Over half of the MAPK genes shown in 5B weren't run in PLUMAGE, ostensibly because the mutations were not recurrent (and therefore less likely to be functional), and of the 8 that were only 3 showed significant changes. It remains unclear then why the authors would focus on mutations that likely do not have any effect.

MINOR COMMENTS:

Page 2- "...we provide functional evidence that 5' UTR mutations in the MAP kinase signaling pathway are associated with early metastasis." First of all there is no functional evidence shown in Fig 5. Also, MAPK mutations were associated with bone metastases not early metastases.

Page 6- How do the number of 5'UTR mutations in the PDXs compare to the number of coding region mutations in those samples?

Page 7- Why did they pick those three genes to validate?

Page 9- It would also be interesting to call out which genes have a large number of mutations, and/or which of the mutations are recurrent in that the EXACT same mutation is seen in multiple samples.

Page 9- "However, 55% of 5' UTR mutations..." Should be Supplemental Fig. 5e

Page 10- The authors call out specific prostate-related genes with 5'UTR mutations, however they never circle back to them later on after the PLUMAGE assay. Do they change expression/translation?

Page 10- It would appear that the authors are using recurrent here (and elsewhere) to specify genes with mutated 5'UTRs in 2 or more patients, but not necessarily mutated at the exact same position. This is not clear and should be clarified.

Page 13- Most of the alterations that had an effect on transcription don't alter a known motif. However, are there any common unidentified motifs among them?

Page 13- "...5' UTR alteration present in a patient specimen..." This is unclear and it sounds as if you are talking about one of the 229 patient samples and not the PDX.

Page 13- Same issue highlighted in the major comments. LuCaP78 doesn't have the mutation yet it has a very similar 3.06 fold change in FGF7 expression.

Page 14- The authors state that these mutations alter transcription/translation by disrupting DNA/RNA binding elements. Seeing as they have not provided any functional data to demonstrate these claims the wording is a bit overstated.

Page 14- Any theories as to why the mutation in the QARS 5' UTR might alter both transcription and translation? There appears to be a motif in Fig 4A...

Page 15- "GSEA analyses showed that 5' UTR mutations in localized prostate cancer affected cell cycle pathways..." " 5' UTR mutations are present in genes which are in cell cycle pathways, the effect on the mutations on the gene or on cell cycle is not shown.

Page 15- Do the mutations in MAPK genes that were tested in PLUMAGE increase MAPK signaling or decrease it?

Page 15- Is there any known association with MAPK signaling and bone mets?

Reviewer #3, expert in bioinformatics (Remarks to the Author):

Overall, this is a significant amount of work used to identify the consequence of mutations in the 5' UTR. The approach is novel and of broad relevance beyond prostate cancer. Further, the findings presented here offer a novel angle towards understanding the molecular underpinnings of mCRPC. While I feel this manuscript warrants publication, I suggest the following comments to potentially further strengthen this manuscript:

(1) The authors indicate that they observe a total of 326 mutations across all five PDXs. It would be beneficial if the authors could elaborate on this large quantity. Are these likely germline. And

how does this impact their overall findings which also include somatic events.

(2) In addition to ribosome profiling, if possible, it would be interesting to incorporate publicly available proteomics data as further evidence.

(3) The authors mention that they identify genes with mutations that previously had unknown mechanisms of activation (FGF7). It would be interesting if they could overlay some of their most interesting and frequent genes with 5' UTR alterations in the context of existing mutational landscapes in clinically localized and mCRPC.

We would like to thank our reviewers for comments regarding our manuscript. We were very happy that reviewer #1 found our work, “clever and interesting.” This was further supported by reviewers #2 and #3 who stated that “this manuscript is well-presented and highly relevant,” and that “the approach is novel and of broad relevance beyond prostate cancer,” respectively. Over the past 5 months we have worked tirelessly to address all the comments raised by the reviewers. Below you will find our point-by-point responses (reviewer’s comments are in black and our rebuttals are in blue).

Response to all reviewers:

We were ecstatic when we received news that our manuscript was invited for a revision at Nature Communications. As you will see, we have addressed all the reviewer’s comments with new data supporting our original claims. Of note, this has resulted in 2 new main figures, along with 4 new supplementary figures. With regards to the requests that we present more data supporting the clinical relevance of functional 5’ UTR mutations, we now demonstrate that patients with 5’ UTR mutations that were positive hits in PLUMAGE also exhibited pathway activation as well as increased sensitivity to taxane-based chemotherapy. We also provide new experiments demonstrating that the 3’ barcodes do not alter gene expression in multiple wild type and mutant construct pairs, and that our method for isolation of ribosome-associated mRNAs did not affect the sensitivity of our method.

We are also delighted to present two new experiments that drive home the functional importance of somatic mutations within the 5’ UTR. As mentioned by reviewer #2, the gold standard for validating functional 5’ UTR mutations would be to create isogenic cells. To this end, we have generated isogenic cell lines against a functional CKS2 5’ UTR mutation we identified by PLUMAGE. CKS2 is a regulatory subunit of cyclin dependent kinases that can promote tumor progression. Using new state-of-the-art CRISPR base-editing technology, we discovered that mutating just one nucleotide within the CKS2 5’ UTR significantly increased protein expression by creating an upstream AUG and N-terminal extension but had no impact on mRNA levels. This experiment is the first to show that a prostate cancer patient-based 5’ UTR mutation is sufficient to alter gene expression of an oncogenic protein.

In a second experiment, we conducted an electrophoretic mobility shift assay (EMSA) using wild-type and mutated FGF7 5’ UTR genomic sequences and observed that the MYC:MAX heterodimer only bound the mutant. Therefore, we provide new functional evidence that a single nucleotide somatic mutation within a 5’ UTR genomic sequence can create new oncogenic transcription factor binding sites.

Overall, we are very grateful for comments provided by the reviewers. They strengthened the core message of our paper and we provide a detailed response to each point below.

Response to Reviewer #1, expert in 5’ UTR mutations:

Reviewer #1 wrote, “In this study, Lim et al. developed a high-throughput method called PLUMAGE to investigate the effects of mutations in 5’ UTR of mRNAs in prostate cancer patients on translation and transcription. By using PLUMAGE, they identified hundreds of

mutations in 5' UTR that can significantly alter translation or transcription levels. Further computational analyses showed that these cancer-associated mutations are mainly located to the cis-elements. Besides, they reported that even a single mutation within cis-elements is sufficient to change translation or transcription levels. Although massively parallel reporter assays have been intensively used to investigate the relationship of sequence contexts to transcription or translation, PLUMAGE provides a new way to study translation and transcription simultaneously, by estimating the relative abundance of 3' end tethered unique sequences. The using of long-read and short-read sequencing technologies to couple the WT and mutated 5' UTR with the unique barcodes is clever and interesting. However, the current version is a bit premature, and some major concerns need to be addressed."

"My major concern is they used the barcodes (8 nt or 30nt) in the 3' end of the reporter to estimate transcription and translation activity, by isolating total RNAs and ribosome associated RNAs. There are a few potential problems of this design."

We thank the reviewer for highlighting the strengths of PLUMAGE stating that our technology is "a new way to study translation and transcription simultaneously," a feature not widely accessible for nearly all massively parallel reporter assays currently available. We were also thrilled that the reviewer found that coupling of long and short-read sequencing was "clever and interesting." We are proud of this technological feature, which has made the PLUMAGE assay possible, and showcases the utility of combining multiple sequencing platform technologies. We have carefully considered all of the reviewers very helpful comments with regards to our barcodes as well as other important concerns which are addressed point-by-point below:

1. Reviewer #1 states, "The random barcodes in 3' UTR may alter translation efficiency as well as mRNA stability. The authors should show evidence that any translation changes associated with 5' UTR mutations they observed are not due to random barcodes in 3'UTR."

This is a very astute comment because it is possible that the barcodes in the 3' end of our constructs can impact mRNA stability and translation. As such, we have completed additional experiments to show that different random 30-bp barcodes do not alter translation efficiency and mRNA stability. Specifically, 4 additional barcodes within plasmids for both NUMA1 and AKT3 (WT and mutant) were tested in luciferase assays, and all showed the corresponding change in translation efficiency by the mutant. These results have been discussed in the manuscript (page 15) and the data can be found in Supplementary Fig. 11a – 11b.

Additionally, using 2 different barcodes for both FGF7 and FOS, we show that transcript levels were not altered by the presence of different randomer barcodes. Therefore, the barcodes are not responsible for the transcriptional changes we observed. To further support this point, using actinomycin D, we show that the change in FGF7 and FOS transcription levels were not due to decreased degradation of FGF7 or FOS mRNA. These results have been discussed in the manuscript (page 14) and the data can be found in Supplementary Fig. 10.

Together, these experiments demonstrate that the 3' end barcodes used in PLUMAGE did not alter the translation or stability of our assays.

2. Reviewer #1 comments, "Change of RNA levels may stem from either transcriptional regulations or post-transcriptional regulations, e.g. RNA stability. Although they reported that

many mutations can be located to cis-elements, suggesting that lots of mutations are likely to alter transcriptional regulations, mutations on RNA stability need to be excluded or normalized.”

We appreciate the reviewer pointing this out, as it is an important concept. To address this, we conducted experiments using actinomycin D, which inhibits the synthesis of new mRNA, thereby allowing the assessment of mRNA decay by measuring mRNA abundance. Actinomycin D was added to PC3 cells in culture for 48 hours following transfection of FGF7 and FOS WT and mutant plasmids. 1 hour later, cells were harvested, and RNA extracted for qPCR analysis. In both FGF7 and FOS, where the mutants resulted in an increase in transcript levels, the rate of mRNA decay were not decreased in the mutant. These results show that the increase in FGF7 and FOS mRNA was not due to less gene-specific mRNA turnover in the mutant setting. These findings have been included in the manuscript (page 14) and the data can be found in Supplementary Fig. 10.

3a. The reviewer notes, “The authors used sucrose gradient to isolate ribosome associated mRNAs. I did not find in method sections which fraction they used, monosome or polysome? Or all the fractions?”

We thank the reviewer for this question. For our polysome analysis we combined all polysomes fractions after the disome as determined by our spectrophotometer readings (Supplementary Fig. 7b). We have now updated this in the text (pages 9, 40 and 41) and have clarified this in the figure as wells (Supplementary Fig. 7b).

3b. Reviewer #2 comments, “I wondered the sensitivity of this method to estimate translational changes. For example, a WT reporter mRNA is translated by monosome, whereas the mutation reporter mRNA is translated by multiple ribosomes, i.e. translation is increased by mutations. However, the two reporter mRNAs are classified into ribosome associated mRNAs, thus may not have significant difference in detected RNA levels.”

This is a very important question. To ensure that we were only measuring mRNAs with multiple associated ribosomes ($n > 2$), we combined polysome fractions after the disome (Supplementary Fig. 7b). Importantly, monosomes were not included in the polysome-associated RNA samples and analysis. We have changed the text to clarify this on page 12.

3c. Reviewer #2 suggests, “They may use a similar method (mean ribosome loading, MRL) in PMID: 31267113 to increase the sensitivity of reporter.”

We agree that the method described by the Seelig Lab is very interesting and robust because it measures mRNA levels in each polysome fraction and uses machine learning to finding trends in how different 100-200 base 5' UTR mRNA sequences impact polysome distribution which they call mean ribosome loading (MRL). This method is different from our methods because we pooled all polysome fractions after the disome.

We chose our method because of the following reasons:

- We were interested in discovering mutations that cause relatively large changes in polysome loading. The benefit of this is that our method increases the likelihood that a positive hit by PLUMAGE is functionally relevant. Therefore, our method increases specificity.
- Furthermore, the MRL method relies on a relative comparison on all fractionation samples

without measuring the total input mRNA. In our opinion in order to obtain an accurate measurement of translation efficiency, it is critical to measure not only polysome associated mRNAs, but also total mRNA levels. This enables reproducible normalization of translation efficiencies across WT and mutant pairs.

- Another reason why we chose to pool polysomes was because this would substantially decrease the cost of our assay compared to MRL measurements. Currently, for each replicate PLUMAGE only requires 3 sequencing libraries (DNA, RNA, polysome-bound mRNA). In comparison an MRL analysis requires 12 sequencing libraries per replicate (one library per fraction, 12 fractions total). Therefore, PLUMAGE decreases sequencing costs by 4-fold, which we hope will bolster its use by the research community at large.
- Lastly, we chose not to use the MRL method because the strength of the Seelig Lab paper was based on the fact that they used machine learning to identify patterns from hundreds of thousands of 100-base 5' UTR sequences. Given that we only measured 914 unique 5' UTR sequences, our significantly smaller dataset was not amenable to their machine learning algorithm which requires a very large test set to validate findings.

However, the comment from reviewer #1 remains important which is how sensitive is PLUMAGE in identifying 5' UTR mutations that impact mRNA translation? To show the sensitivity of PLUMAGE we conducted an orthogonal test of our method. In particular, we have now isolated and sequenced the 80S fraction from the PLUMAGE library to determine if we can observe a shift from polysome to monosome or vice-versa in 5' UTR mutations that cause changes in translation efficiency. Indeed, taking polysome-bound mRNA as a ratio of 80S-bound mRNA, we demonstrate a strong correlation with polysome/total mRNA (Pearson $r = 0.78$, $p = 0.0001$), suggesting that the translation changes that we originally observed reflect differential ribosome loading and occupancy. This result is now described in the main manuscript (page 13), and the data can be found in Supplementary Fig. 9a.

Review #1, other concerns.

1. Reviewer #1 states, "Figure 1g. The three mutations were tested to demonstrate results from PLUMAGE are consistent with the results from ribosome profiling (Fig. 1d). It would be better if a transcriptome wide correlation between PLUMAGE and ribosome profiling is shown."

We thank the reviewer for this comment. We agree that a transcriptome wide comparison between our PLUMAGE data and ribosome profiling data from PDX models would be a good analysis. However, very few mutations that were identified by ribosome profiling were included in our PLUMAGE libraries ($n = 16$), which primary focused on patient-based mutations (Fig. 2), thereby significantly limited the power of the study. Of the 16 that were present in both assays, 50% demonstrated a change in gene expression in the same direction. However, 50% did not. There are two potential reasons for this difference. First, the ribosome profiling was conducted in PDX tissues which by definition are heterogenous and contain not only tumor, but also other normal cellular components that can impact gene expression readouts such as mRNA levels and translation efficiency. Second, the ribosome profiling findings were only an association between somatic 5' UTR mutations and differences in transcript or translation levels in PDX tissues. It is very possible that in PDX tumors there are other causes of gene expression changes besides the 5' UTR mutation. For both of these reasons, we emphasized the need to conduct orthogonal assays in our manuscript. For example, we utilized gene-specific qPCR (Figs. 1d, 4b, Supplementary Fig. 10b), luciferase assays (Figs. 1d, 5b, Supplementary Figs. 11a and 11b), actinomycin D experiments (Supplementary Fig. 10a), an electrophoretic mobility

shift assay (Fig. 4e), and base-editing of an endogenous 5' UTR (Fig. 6) to determine the robustness of PLUMAGE and the functionality of distinct 5' UTR mutations.

2. Reviewer #1 asks, "Figure 2e. Are the mutations that significantly alter translation efficiency enriched in the TOP mRNA?"

There were 5 mutations that affect 5' TOP motifs in our analysis, and none of those were observed to significantly alter translation efficiency by PLUMAGE.

3. Reviewer #1 comments, "Figure 3c and 3d. Although the correlation between cell lines remains high, the significantly reduced correlation indicates that the effect of some mutations could be cell line dependent."

We agree with the reviewer that there is a reduction in correlation between the 293T and PC3 cell lines which suggests that for some genes, context matters. However, the correlation between cell lines is still very robust with a Pearson $r > 0.88$ ($p < 0.0001$) suggesting that context specificity is limited to a small number of genes. We have now updated our manuscript with a discussion about cell line differences on page 13.

4. The reviewer asks, "Are the flanking sequences surrounded the mutations that significantly alter transcription or translation activity subjected to negative natural selection, compared with neutral mutations?"

This is a very interesting question because it has been previously reported that germline variants within the 5' UTR that create *cis*-regulatory elements such as uORFs are under strong negative selection (Whiffin N et al. Nat Com 2020). To address this point within our dataset, we asked if our significant PLUMAGE hits ($n = 190$, $FDR < 0.1$) were associated with structural variations in sequences flanking the mutation compared to hits that did not show a significant change in gene expression in PLUMAGE ($n = 355$, $FDR > 0.1$). Comparing the per-patient localized copy number for each mutation, we found no increase in copy number losses in functional 5' UTR mutations (page 13, Supplementary Fig. 9b). This result is tempered by the small size of our dataset and additional work is required to determine how applicable our findings are to the entire genome.

Response to Reviewer #2, expert in prostate cancer genomics:

SUMMARY:

Reviewer #2 wrote, "In this study Lim et al. have examined mutations in the 5' UTR in 5 PDX models of prostate cancer and 229 patient samples from both localized and mCRPC. They characterized the gene families affected by these mutations and also their potential to disrupt known DNA and RNA binding motifs. They then examined the effect of these mutations on transcription and translation via a novel high-throughput technique they named PLUMAGE and validated some candidates orthogonally. Lastly, they described mutations in MAPK pathway genes which are only seen in mCRPC."

"The work in this manuscript is well-presented and highly relevant. There are still major gaps in knowledge about how alterations in non-coding regions of the genome, including UTRs, affect

gene expression. As such, their findings would be of interest to many researchers not just those in the prostate cancer field.”

We thank the reviewer for highlighting the relevance of our work in helping to bridge the knowledge gap in “how alterations in non-coding regions of the genome, including UTRs, affect gene expression,” extending the relevance of our work to many researchers outside of the prostate cancer field.

Moreover, the reviewer states, “Particularly novel is their new high-throughput method PLUMAGE which allows for the simultaneous measurement of the effect of mutations on expression AND translation.”

We thank the reviewer for recognizing the novelty of our new method, which we believe is a unique strength of this manuscript and will be of great interest to Nature Communications’ wide audience.

Reviewer #2 goes on to say, “Aside from specific major and minor comments delineated below, the manuscript would be improved by focusing more on the positive results that came out of the PLUMAGE screen of mutations. Functional experiments manipulating these sites by creating isogenic cell lines would be the gold standard, and would allow for the determination of the effects in the native context of that transcript’s stability. However, even in the absence of further functional data it would be satisfying to have more emphasis on the mutations that result in significant changes: what cancer-related pathways would they regulate and in what direction, is there anything that delineates them from the non-significant ones, is there a difference in localized vs CRPC, are they specifically associated with any clinical parameters? As written the authors answer many of these questions only for their list of overall mutations, despite the fact that their unique analysis with PLUMAGE is what truly differentiates their study.”

We thank the reviewer for these very insightful comments, and we fully agree that the gold standard functional validation experiment would be to create isogenic cell lines that test whether 5’ UTR mutations are indeed functional in their endogenous context. In response to this comment, we have engineered a somatic 5’ UTR mutation into the oncogenic *CKS2* gene, which increased mRNA translation in the PLUMAGE screen (Fig. 6), using CRISPR-mediated base editing. We now demonstrate that this endogenous mutation introduces an upstream in-frame AUG which increased the translation of *CKS2* without any change to mRNA levels. This observation corroborates our PLUMAGE findings and demonstrates that 5’ UTR mutations can coordinately impact mRNA translation by altering RNA-based *cis*-regulatory elements in their endogenous context. The new results are described in the main manuscript (pages 16 - 17), and data are shown in Fig. 6.

The reviewer also asked us to determine if our positive PLUMAGE hits were associated with pathway changes or specific clinical features. To determine if functional 5’ UTR mutations of MAP kinase regulators can impact pathway-specific gene expression, we analyzed RNA sequencing data from patients and PDX models harboring MAP kinase pathway mutations we tested in PLUMAGE. Interestingly, 3 patients with functional MAP kinase pathway mutations to *FOS*, *FGF7* and *MECOM* that were predicted to increase signaling by PLUMAGE demonstrated upregulation of a RAS-driven prostate cancer MAP kinase pathway gene signature (Fig. 7b) (Wang J et al. Cancer Research 2012). Interestingly, 3 patients with non-functional 5’ UTR mutations to MAP kinase pathway genes *PTPN7*, *JUN* and *DDIT3* did not exhibit changes in

gene expression (Fig. 7b). These findings demonstrate that functional 5' UTR mutations are associated with alterations in oncogenic pathway activity and are discussed on page 18.

In addition, we have now conducted a thorough study of how functional 5' UTR mutations of MAP kinase regulators correlate with patient outcomes. We analyzed multiple patient endpoints including progression free survival, overall survival, time to metastases, Gleason score, and duration on therapeutic agents. Interestingly, we observed that patients with functional MAP kinase pathway mutations as determined by PLUMAGE (FDR < 0.1) were more likely to have a sustain response to the microtubule inhibitor and chemotherapy Taxotere compared to patients without functional mutations (Supplementary Fig. 12). This is consistent with a previous report which demonstrated that taxane-mediated apoptosis is dependent on MAP kinase signaling (Bacus SS et al. Oncogene 2001).

Together, our new findings demonstrate that functional 5' UTR mutations identified by PLUMAGE can impact endogenous gene expression and are associated with MAP kinase pathway activation and response to chemotherapy.

MAJOR COMMENTS:

1) Reviewer #2 comments, "Figure 1C and 4C- In these two panels the authors are trying to associate changes in the gene expression or translation to the presence of the 5' UTR mutations. In order to do so they compare expression or TE in each line to 5 normal prostate samples. These comparisons are flawed in that it would appear that a large percentage of differences that are picked up as significant are actually just demonstrating PCa vs normal instead of differences due to any mutations. For example, as highlighted in Fig 1C, a mutation in ADAM32 in LuCaP147 is associated with decreased TE in that sample. However, in Supplementary Table 1C they report that TE is also decreased in 3 of the other 4 LuCaP PDX despite the fact that they do not have the 5'UTR mutation. These issues persist with other "significant" changes - in LuCaP78 3 genes are reported to have mutations that alter TE, but 2 of these genes show altered TE in other PDX that don't have the mutation. In LuCaP145 all of the TE changes linked to mutations are seen in other PDX. The authors should address and provide any statistical support that would confirm mutation association with TE."

We thank the reviewer for this astute observation. We agree that some genes without 5' UTR mutations also exhibited differences at the mRNA or translation efficiency levels in our PDX tissue-based studies. This makes correlating distinct 5' UTR mutations with genetic outputs from PDX specimens very difficult. Indeed, even without a 5' UTR mutation, it is possible that transcription or translation of a specific gene is altered by other mechanisms in cancer. Furthermore, despite computationally subtracting mouse RNA, each PDX contains a plethora of non-cancer tissues which can impact the interpretation of this type of tissue-based study. Given the intrinsic variability of our tissue-based analysis we found no additional statistical measurement that could prove without a doubt that a specific mutation caused (or did not cause) the changes in gene expression observed in the PDX models. This is why we were compelled to use multiple orthogonal assays to determine the functionality of 5' UTR mutations. This included qPCR analysis (Fig. 4b, Supplementary Fig. 10), luciferase assays (Figs. 1d, 5b, Supplementary Figs. 11a and 11b), CRISPR-mediated base editing (Fig. 6), and the development of the PLUMAGE technology (Figs. 1e, 1g, and 3). All of these orthogonal assays serve to validate the functionality of distinct somatic 5' UTR mutations and the associations we see in tissues.

Nonetheless, to be transparent to our readers and rigorous in our approach, we have now included a discussion about the interpretation of the PDX study (page 7).

2) The reviewer notes, “Figure 5- After having run PLUMAGE to find which of the 5’UTR mutations result in functional changes in expression or translation the authors now go back to their original list of all the mutations identified. This is especially confusing given that the PLUMAGE analysis showed that only about a third of the mutations caused any significant change. Over half of the MAPK genes shown in 5B weren’t run in PLUMAGE, ostensibly because the mutations were not recurrent (and therefore less likely to be functional), and of the 8 that were only 3 showed significant changes. It remains unclear then why the authors would focus on mutations that likely do not have any effect.”

This is a fair point, and we appreciate the comment. We have now conducted two additional studies of our MAP kinase pathway 5’ UTR mutations that were predicted to increase pathway activity by PLUMAGE.

First, we compared RNAseq data from patients or PDX models with functional 5’ UTR mutations to the MAP kinase signaling pathway (PLUMAGE FDR < 0.1) to those with non-functional mutations (PLUMAGE FDR > 0.1). Interestingly, 3 patients with functional MAP kinase pathway mutations to FOS, FGF7 and MECOM that were predicted to increase signaling demonstrated upregulation of a RAS-driven prostate cancer MAP kinase pathway gene signature (Fig. 7b) (Wang J et al. Cancer Research 2012). On the contrary, 3 patients with non-functional 5’ UTR mutations to MAP kinase pathway genes PTPN7, JUN and DDIT3 did not exhibit changes in gene expression (Fig. 7b). These findings demonstrate that functional 5’ UTR mutations have the ability to impact cancer-associate pathway activity and are now discussed on page 18.

Second, we conducted a thorough study of how functional MAP kinase 5’ UTR mutations correlate with patient outcomes. We observed that patients with functional MAP kinase pathway mutations (FDR < 0.1) that are predicted to increase signal by PLUMAGE (FOS and MECOM) were more likely to have a sustain response to taxane-based chemotherapy compared to patients with nonfunctional mutations (Supplementary Fig. 12). This is consistent with a previous report which demonstrated that taxane-mediated apoptosis is dependent on MAP kinase signaling (Bacus SS et al. Oncogene 2001). These findings are discussed on page 18.

Together these studies demonstrate that functional 5’ UTR mutations can associate with pathway activity as well as patient outcomes.

MINOR COMMENTS:

1) Reviewer #2 writes, “Page 2- “...we provide functional evidence that 5’ UTR mutations in the MAP kinase signaling pathway are associated with early metastasis.” First of all there is no functional evidence shown in Fig 5. Also, MAPK mutations were associated with bone metastases not early metastases.”

We thank the reviewer for this comment. Based on our new experiments we have changed the sentence to “Using gene editing technology, we validate that a single point mutation in the oncogenic CKS2 5’ UTR can increase mRNA specific translation. Turning to a molecular pathway critical for cancer, we provide evidence that functional 5’ UTR mutations in the MAP

kinase signaling pathway can upregulate pathway-specific gene expression and are associated with distinct clinical outcomes.”

2) Reviewer #2 asks, “Page 6- How do the number of 5’UTR mutations in the PDXs compare to the number of coding region mutations in those samples?”

In general, we found more mutations within the coding regions at an absolute and per megabase level similar to what we observed in the human mCRPC specimens (see below, Fig. 2a and Supplementary Fig. 5d).

3) Reviewer #2 asks, “Page 7- Why did they pick those three genes to validate?”

We were interested in validating genes that had 5’ UTR mutations, exhibited changes in gene expression and were also of relevance in cancer. ADAM32 plays an important role in many biological processes including fertilization, tumor development, and inflammation (PMID: 27391163, 12568724). Polymorphisms of COMT have been associated with sporadic prostatic carcinogenesis (PMID: 16492910). Therefore, these genes were of interest in prostate cancer. ZCCHC7 is associated with RNA degradation and was chosen as a control since its 5’ UTR mutation that did not impact gene expression (PMID: 27546437).

4) In addition, reviewer #2 suggests, “Page 9- It would also be interesting to call out which genes have a large number of mutations, and/or which of the mutations are recurrent in that the EXACT same mutation is seen in multiple samples.”

This is a good idea. To this end, we have now included Supplementary Table 4g to highlight which mutations are recurrent in the exact same genomic location. In particular, IKZF4 had the largest number of recurrent mutations in the exact same genomic location (14 SNVs), followed

by FAM160A1, CHRFAM7A and PROX1. The genes with at least 3 or more recurrent mutations include EVI2A, PROX1 and DDX11. Several cancer-associated genes such as MECOM, NOTCH2, CDH12 and FTH1 were also found to have 3 or more mutations.

5) Reviewer #2 pointed out, “Page 9- “However, 55% of 5’ UTR mutations...” Should be Supplemental Fig. 5e”

Thank you for catching this. We have updated the text.

6) The reviewer states, “Page 10- The authors call out specific prostate-related genes with 5’UTR mutations, however they never circle back to them later on after the PLUMAGE assay. Do they change expression/translation?”

We have now circled back to two of these genes that validated in PLUMAGE including the oncogenic FTH1 and tumor suppressive MECOM on page 13.

7) The reviewer comments, “Page 10- It would appear that the authors are using recurrent here (and elsewhere) to specify genes with mutated 5’UTRs in 2 or more patients, but not necessarily mutated at the exact same position. This is not clear and should be clarified.”

The majority of mutations that are recurrent are altered in the same gene and not in the same position. In order to clarify this, we have now added a table to show all recurrent mutations at the exact same genomic location (Supplementary Table 4g).

8) The reviewer asks, “Page 13- Most of the alterations that had an effect on transcription don’t alter a known motif. However, are there any common unidentified motifs among them?”

To address this question, we conducted a MEME analysis for highly conserved motif-like sequences within our positive transcriptional PLUMAGE hits. We found no significantly enriched motifs. It is possible that these mutations impact structural features of the DNA, which cannot be determined by a motif-based analysis.

9) Reviewer #2 comments, “Page 13- “...5’ UTR alteration present in a patient specimen...” This is unclear and it sounds as if you are talking about one of the 229 patient samples and not the PDX.”

Thank you for noticing this. We have edited the text accordingly.

10) The reviewer states, “Page 13- Same issue highlighted in the major comments. LuCaP78 doesn’t have the mutation yet it has a very similar 3.06 fold change in FGF7 expression.”

This is an important comment which we have addressed above in major comment #1.

11) The reviewer notes, “Page 14- The authors state that these mutations alter transcription/translation by disrupting DNA/RNA binding elements. Seeing as they have not provided any functional data to demonstrate these claims the wording is a bit overstated.”

To address this comment, we now provide functional evidence that mutating the genomic sequence of the FGF7 5’ UTR can enable MYC/MAX binding (Fig. 4e). Furthermore, we

demonstrate that the introduction of an upstream AUG in its endogenous context can increase the translation of oncogenic CKS2 (Fig. 6). Collectively, these data provide direct functional evidence that disrupting *cis*-regulatory elements can impact gene expression at the transcript or translation levels.

12) Reviewer #2 asks, “Page 14- Any theories as to why the mutation in the QARS 5' UTR might alter both transcription and translation? There appears to be a motif in Fig 4A...”

The decrease in QARS transcript levels may be secondary to a mutation in the ELK4 DNA binding motif (Supplementary Table 5a). However, at the RNA level, the mutation does not affect any known binding motifs. It is tempting to speculate that the change in post-transcriptional gene expression is due to structural alterations of the 5' UTR, which have been shown to impact mRNA translation (Schuster et al. Trends in Cancer 2019).

13) The reviewer states, “Page 15- “GSEA analyses showed that 5' UTR mutations in localized prostate cancer affected cell cycle pathways... “ 5' UTR mutations are present in genes which are in cell cycle pathways, the effect on the mutations on the gene or on cell cycle is not shown.”

The reviewer is correct. GSEA analyses are intended to demonstrate pathway enrichment which does not equate to functionality. We have now revised the sentence to say, “Indeed, GSEA analyses showed that 5' UTR mutations in localized prostate cancer enrich for cell cycle pathways, whereas mutated genes in mCRPC enrich for metabolism and the MAP kinase signaling pathway (Pages 17 - 18).”

14) Reviewer #2 asks, “Page 15- Do the mutations in MAPK genes that were tested in PLUMAGE increase MAPK signaling or decrease it?”

This is an important question. We now provide evidence that positive PLUMAGE hits that are predicted to increase MAP kinase activity are associated with an increase in pathway activation as determined by RNAseq of the patient/PDX specimens that possess these mutations (Fig. 7b). This is discussed on page 18.

15) Lastly, reviewer #2 asks, “Page 15- Is there any known association with MAPK signaling and bone mets?”

Yes, it has been recently shown by the Abate-Shen Lab that MAP kinase pathway hyperactivity drives bone metastases in murine models and is associated with metastases in patients with castration resistant prostate cancer (Arriaga et al. Nature Cancer 2020).

Reviewer #3, expert in bioinformatics:

Reviewer #3 comments, “Overall, this is a significant amount of work used to identify the consequence of mutations in the 5' UTR. The approach is novel and of broad relevance beyond prostate cancer. Further, the findings presented here offer a novel angle towards understanding the molecular underpinnings of mCRPC. While I feel this manuscript warrants publication, I suggest the following comments to potentially further strengthen this manuscript.”

We thank the reviewer for his/her enthusiasm for our work, and for pointing out the novelty and broad relevance beyond prostate cancer.

1) Reviewer #3 states, “The authors indicate that they observe a total of 326 mutations across all five PDXs. It would be beneficial if the authors could elaborate on this large quantity. Are these likely germline. And how does this impact their overall findings which also include somatic events.”

Thank you for this comment. We agree with the reviewer that 326 mutations are a large number of alterations for 5 PDX prostate tumor specimens. The explanation for this is that one tumor (LuCaP 147) was hypermutated and contributed the majority of the mutations observed. This has been clarified in the text on page 6 (see Supplementary Table 1a). These mutations were somatic and not germline because we compared tumor to matched normal tissues. Therefore, it is unlikely that germline alterations contribute to the 326 mutations we identified in these PDX specimens.

2) Reviewer #3 requests, “In addition to ribosome profiling, if possible, it would be interesting to incorporate publicly available proteomics data as further evidence.”

Thank you very much for this suggestion. We considering using the proteomics data from Sinha A et al. Cancer Cell 2019 which was derived from the tissues analyzed for the localized prostate cancer 5' UTR genomic analysis in Figure 2. However, after careful consideration, we were concerned that it would be difficult to make strong associations given that the cancer tissues are heterogeneous, and there are no matched normal controls to determine if a 5' UTR mutation is associated with a change in protein levels.

In order to circumvent these issues, we have performed Tandem Mass Tag (TMT) mass spectrometry on the 5 PDX tissues and 5 normal prostate tissues that were used for 5' UTR genomic sequencing, transcriptome analysis and ribosome profiling in figure 1. Of the 5' UTR mutations in the PDXs that showed a significant change by RNAseq and ribosome profiling, 41 were detected as proteins by TMT-mass spec, and out of these, 28 showed fold change differences (PDX/normal) in the same direction as the ribosome profiling data. In addition, to more directly investigate if 5' UTR mutations lead to changes in protein expression, we also engineered the 5' UTR mutation of CKS2 into a cell line and showed an increase in protein expression of endogenous CKS2 (Fig. 6). Therefore, as reviewer #3 suspected, 5' UTR mutations are associated with and can lead to changes in protein levels in tissue specimens and in cells.

3) Lastly, the reviewer notes, “The authors mention that they identify genes with mutations that previously had unknown mechanisms of activation (FGF7). It would be interesting if they could overlay some of their most interesting and frequent genes with 5' UTR alterations in the context of existing mutational landscapes in clinically localized and mCRPC.”

This is a good point. We have now analyzed the Quigley et al. Cell 2018 metastatic castration resistant prostate cancer (mCRPC) dataset, to determine if mutations found in our localized and metastatic patient cohort are also present in another dataset. The Quigley dataset consists of whole genome sequencing of mCRPC patients and was therefore relevant to querying the untranslated genome. When comparing all genes that had 5' UTR mutations in our dataset and

the Quigley dataset, we found a significant overlap of 246 genes (see below). This data has also been included in Supplementary Fig. 5f and mentioned in the manuscript (Page 10).

Interestingly, we also observed that MAP kinase pathway genes with 5' UTR mutations (for example, FGF1 and MAP2K3) were found in both datasets. These findings show that 5' UTR mutations are pervasive in advanced human prostate cancer.

REVIEWERS' COMMENTS

Reviewer #1 (Remarks to the Author):

In this revised manuscript, the authors did a nice job in addressing reviewers' concerns. With additional experiments and convincing results, the manuscript has been much improved. This is a beautiful work and I support the acceptance of the manuscript.

Reviewer #2 (Remarks to the Author):

The authors have done an excellent job in this detailed revision. I am satisfied that all my questions were adequately addressed.

Reviewer #3 (Remarks to the Author):

The authors have addressed my concerns.

Lim et al. Nat Comm – NCOMMS-20-31871A
Point-by-point rebuttal

We are very pleased with the reviewers' responses. Our responses are documented below in blue.

Reviewer #1 (Remarks to the Author):

In this revised manuscript, the authors did a nice job in addressing reviewers' concerns. With additional experiments and convincing results, the manuscript has been much improved. This is a beautiful work and I support the acceptance of the manuscript.

We thank reviewer #1 for his/her comments. We agree that the revision process has strengthened the final paper.

Reviewer #2 (Remarks to the Author):

The authors have done an excellent job in this detailed revision. I am satisfied that all my questions were adequately addressed.

We are very thankful for his/her comments.

Reviewer #3 (Remarks to the Author):

The authors have addressed my concerns.

We thank reviewer #3 for his/her thoughtful comments.